# Participation of Extracellular Vesicles from Zika-Virus-Infected Mosquito Cells in the Modification of Naïve Cells’ Behavior by Mediating Cell-to-Cell Transmission of Viral Elements

**DOI:** 10.3390/cells9010123

**Published:** 2020-01-04

**Authors:** Pedro Pablo Martínez-Rojas, Elizabeth Quiroz-García, Verónica Monroy-Martínez, Lourdes Teresa Agredano-Moreno, Luis Felipe Jiménez-García, Blanca H. Ruiz-Ordaz

**Affiliations:** 1Departamento de Biología Molecular y Biotecnología, Instituto de Investigaciones Biomédicas, Universidad Nacional Autónoma de México, Av. Universidad 3000, Ciudad Universitaria, Coyoacán, Ciudad de México 04510, México; pedropablo.martinezrojas@gmail.com (P.P.M.-R.); geliqg@gmail.com (E.Q.-G.); vmonroy@iibiomedicas.unam.mx (V.M.-M.); 2Departamento de Biología Celular, Facultad de Ciencias, Universidad Nacional Autónoma de México, Av. Universidad 3000, Ciudad Universitaria, Coyoacán, Ciudad de México 04510, México; agredano-moreno@ciencias.unam.mx (L.T.A.-M.); luisfelipe_jimenez@ciencias.unam.mx (L.F.J.-G.)

**Keywords:** extracellular vesicles, Zika virus, cellular communication, C6/36 cells, human monocytes, endothelial vascular cells

## Abstract

To date, no safe vaccine or antivirals for Zika virus (ZIKV) infection have been found. The pathogenesis of severe Zika, where host and viral factors participate, remains unclear. For the control of Zika, it is important to understand how ZIKV interacts with different host cells. Knowledge of the targeted cellular pathways which allow ZIKV to productively replicate and/or establish prolonged viral persistence contributes to novel vaccines and therapies. Monocytes and endothelial vascular cells are the main ZIKV targets. During the infection process, cells are capable of releasing extracellular vesicles (EVs). EVs are mediators of intercellular communication. We found that mosquito EVs released from ZIKV-infected (C6/36) cells carry viral RNA and ZIKV-E protein and are able to infect and activate naïve mosquito and mammalian cells. ZIKV C6/36 EVs promote the differentiation of naïve monocytes and induce a pro-inflammatory state with tumor necrosis factor-alpha (TNF-α) mRNA expression. ZIKV C6/36 EVs participate in endothelial vascular cell damage by inducing coagulation (TF) and inflammation (PAR-1) receptors at the endothelial surface of the cell membranes and promote a pro-inflammatory state with increased endothelial permeability. These data suggest that ZIKV C6/36 EVs may contribute to the pathogenesis of ZIKV infection in human hosts.

## 1. Introduction

Zika virus (ZIKV) is an emerging arthropod-borne *Flavivirus*, transmitted mainly by mosquitoes of the genus *Aedes*, but the ZIKV infection could also be produced by sexual contact or vertical transmission from mother to child [1,2]. ZIKV was first isolated in 1947 from the blood of a sentinel Rhesus monkey No. 766, stationed in the Zika forest in Uganda. Again, in 1948, ZIKV was isolated in the same forest from a pool of *Aedes (Ae) africanus* mosquitoes. Thereafter, serological and entomological data indicated that ZIKV circulates actively in East and West Africa and South-East Asia. In 2007, ZIKV caused an outbreak of relatively mild disease characterized by rash, arthralgia, and conjunctivitis on Yap Island in the Southwestern Pacific Ocean. This was the first time that the virus was detected outside of Africa [3]. Later, a ZIKV epidemic in Brazil was present in 2015 and spread rapidly throughout South and Central America in 2016. The Pan American Health Organization (PAHO) has received reports of more than 7.5 × 10^5^ cases of Zika in 84 cities or territories in America [3,4]. The ZIKV infection during pregnancy can cause fetal loss, microcephaly, and other brain abnormalities that are classified as congenital Zika syndrome [5,6]. Further, severe forms of encephalopathies, meningoencephalitis, myelitis, uveitis, autoimmunity (Guillain-Barré syndrome), and severe thrombocytopenia have been associated with ZIKV infection [7,8]. The pathogenic mechanisms that give rise to severe forms of Zika are still unclear, and to date, no safe vaccine or specific antiviral treatments for ZIKV infection are available [9]. A rapid and successful expansion of ZIKV has occurred due to the high virulence of circulating strains, immunologically susceptible populations, and the wide distribution of its vectors [10,11].

*Ae. aegypti* and *Ae. albopictus* mosquitoes are the primary vectors of several *Flavivirus* such as ZIKV and dengue virus (DENV) [12]. Female mosquitoes acquire the virus from an infected host during feeding, it undergoes replication in the gut and disseminates to the salivary glands, and the virus is released into the saliva, where it is transmitted to the host during subsequent feeding [13,14]. Cime et al. (2015) reported that *Ae. aegypti* saliva plays an important role during DENV transmission to the host cells. Likewise, they detected an enhanced viral infection of mammalian cells in the presence of mosquito salivary gland extract [15]. However, the mechanisms in the transmission of *Flavivirus* from vector to host are not entirely understood [16]. In human hosts, monocytes, macrophages, endothelial vascular cells, and central nervous system cells are identified as main ZIKV target cells [17,18,19]. During differentiation or activation, cells release extracellular vesicles (EVs) [20]. EVs are considered crucial mediators of intercellular communication and play a role in the pathophysiology of inflammation-associated disorders [21].

EVs are a heterogeneous group of particles naturally released by the cells, delimited by a lipid bilayer, and cannot replicate. The classification proposed by the International Society of Extracellular Vesicles (ISEV) has established that EVs can be distinguished by their biogenesis. Vesicles are derived from the plasma membrane (microparticles [MPs]) and are also derived from endosomal maturation (exosomes). Further, they differ in size, where the MPs (> 200 nm) are grouped as large EVs (lEVs), and the exosomes (< 200 nm) are grouped as small EVs (sEVs) [22]. These EVs can be identified by the presence of different membrane markers (phosphatidylserine [PS] in lEVs or tetraspanins in sEVs) or by their internal content, since they transport active biomolecules (proteins and different types of RNA) capable of modifying the response of the cells with which they interact [22,23].

Small EVs are formed as intraluminal vesicles within multivesicular bodies during the endosome maturation process and released into the extracellular space through highly specialized cellular secretory pathways [24]. During the infectious process by some RNA viruses such as flaviviruses, the viral replication cycle and the biogenesis of sEVs can converge, so different viral components (antigens, genomes, or complete viruses) can be part of the internal content, being potential vehicles for viral transmission, evasion of the host’s immune response, and the enhancement of pathophysiological processes by promoting the spread of the pathogen to immunologically privileged sites [25,26]. Therefore, sEVs are considered a new, alternative mechanism that is efficient for viral spread [27]. Large EVs are formed by cytoskeleton rearrangement and released from the plasma membrane after the cell activation process [24]. In blood circulation, MPs facilitate cell–cell interaction and induce different responses associated with inflammation, thrombosis, or vascular dysfunction [28]. Virus-infected cells secrete lEVs that may contain viral proteins and RNAs [29]. Little is known about the EV participation function in the vector–human host interaction during the flaviviruses transmission-infection processes. Recently, Vora et al. (2018) reported that DENV-infected mosquito cells release EVs that contain infectious DENV RNA and proteins, favoring viral transmission from the vector to human keratinocytes and endothelial cells (ECs) [16]. Likewise, Reyes-Ruiz et al. (2019) reported that sEVs from DENV-infected mosquito cells have homologous proteins to human CD9 tetraspanin, containing virion-like particles inside them [30].

To date, the participation of EVs released from *Aedes* mosquito cells during the ZIKV infection process has not been described. This study aimed to evaluate the potential role of EVs from ZIKV-infected C6/36 cells in viral-element, cell-to-cell transmission to the main host’s target cells (monocytes and endothelial vascular cells) as well in the naïve cellular behavior modification. EVs from ZIKV-infected mosquito cells were then isolated by differential ultracentrifugation, characterized by nanoparticle tracking analysis, identified by transmission electron microscopy, and subject to phosphatidylserine (MPs) and tetraspanin CD63 (exosomes) detection by cytofluorometry assays. The isolated sEVs were purified by using paramagnetic beads coated with anti-CD63 antibodies, thus demonstrating their endosomal origin. The possible modification of cellular behavior mediated by ZIKV C6/36 EVs was evaluated by using different cell activation assays. Our results support that EVs (small and large) from ZIKV-infected *Aedes* mosquito cells modify host cells responses, which could be implicated in the pathogenic mechanisms associated with the progression to severe forms of the disease.

## 2. Materials and Methods

### 2.1. Cell Cultures and Zika Virus Strain

Larvae lysate cells (C6/36) from mosquitos *Aedes albopictus* (ATCC CRL-1660, USA), kidney epithelial cells (Vero) from monkeys *Cercopithecus aethiops* (ATCC CCL-81), human monocytes (THP-1) from peripheral blood (ATCC TIB-202), and human endothelial cells (HMEC-1) from the dermal microvasculature (ATCC CRL-3243) were used in this study. The C6/36 cells were maintained in Leibovitz L15 medium (Biowest, Riverside, MO, USA) supplemented with 10% (*v*/*v*) fetal bovine serum (FBS; Biowest, Nuaillé, France), 10% tryptose phosphate broth (DIFCO, Lawrence, KS, USA), 1% 200 mM L-glutamine (Biowest), and 1% antibiotic solution (10,000 U/mL penicillin, 10 mg/mL streptomycin, and 25 μg/mL amphotericin B; Biological Industries, Cromwell, CT, USA) and were incubated at 28 °C without CO_2_ (Lab-Line Ambi-Hi-Low Chamber, Lab-Line Instruments Inc., Melrose Park, IL, USA). The Vero and THP-1 cells were maintained in Dulbecco’s Modified Eagle medium (DMEM, Biowest) and Roswell Park Memorial Institute (RPMI) 1640 medium (Biowest), respectively. Both media were supplemented with 10% Fetal Bovine Serum (FBS), 1% 200 mM L-glutamine, and 1% antibiotic solution. The HMEC-1 cells were maintained in MCDB-131 medium (Sigma-Aldrich, St. Louis, MO, USA) supplemented with 10% FBS, 1% 200 mM L-glutamine, 10 ng/mL epidermal growth factor (Sigma-Aldrich), 1 µg/mL hydrocortisone (Sigma-Aldrich), and 1% antibiotic solution. The Vero, THP-1, and HMEC-1 cells were incubated at 37 °C in 5% CO_2_ (Series 8000 WJ CO_2_ Incubator, Thermo Fisher Scientific, Waltham, MA, USA). Dr. Amadou A. Sall, from Institut Pasteur Dakar, kindly provided the reference strain ZIKV MR766 (Genebank Accession HQ234498.1), with the following passage history: 146 × in suckling mouse, 1 × C6/36 cells, and 1 × Vero cells. Additionally, in our laboratory, it was passaged twice in C6/36 cells.

### 2.2. ZIKV Propagation and Titration

The ZIKV MR766 strain with passage (P) number 3 was propagated in C6/36 cells and titrated in confluent Vero monolayers and was used in all experiments. The ZIKV titer was determined by the lytic plaque assay as follows: Vero cells were seeded into a 24-well culture plate (Corning, Corning, NY, USA) and incubated at 37 °C in 5% CO_2_ until confluence. Each culture well was inoculated with 400 μL of diluted ZIKV (serial log (10-fold) dilutions in serum-free medium with dilution factors from 10^−1^ up to 10^−22^) in duplicate and incubated for 2 h at 37 °C in 5% CO_2_. The viral inoculum was removed, and each well was washed with Phosphate Buffer Saline (PBS) 1× (pH 7.4). The monolayers were overlaid with 1 mL of DMEM medium containing 1% carboxymethylcellulose (Sigma, USA) and 2.5% FBS and afterwards incubated at 37 °C in 5% CO_2_. The plaque formation began to be observed at the 7th day post-infection (PI). Cells were fixed with 96% methanol (J.T.Baker, Fisher Scientific, Allentown, PA, USA) on the 14th day PI and stained with 1% crystal violet (Sigma) for 15 min. The titer was calculated and expressed as plaque-forming units (PFU) per milliliter (mL), according to the following formula: PFU/mL = N/(*V* × *D*), where *N* corresponds to the average number of plaques counted, *V* is the inoculated volume of the viral dilution, and *D* is the less concentrated dilution from which the plaques were counted.

### 2.3. Preparation of Fetal Bovine Serum Depleted of Extracellular Vesicles (EVs) 

The FBS was collected in sterile conical tubes (Corning) and centrifuged (GH3.8 rotor, Beckman GPR Centrifuge; Beckman Coulter, Inc., Brea, CA, USA) at 900× *g* for 10 min and filtered using a 0.22 μm pore (Millipore, Burlington, MA, USA). The samples were then centrifuged (SW28 rotor, Beckman XL-90 Ultracentrifuge, Beckman Coulter, Inc.) at 120,000× *g* for 18 h at 4 °C [22] and stored at 2 °C until use.

### 2.4. ZIKV Infection Assay 

Cell cultures (mosquito C6/36, human monocytes, or endothelial vascular cells) were seeded in 12-well plate culture (Corning) until confluence. They were then infected with ZIKV at a multiplicity of infection (MOI) of 1, as previously described [31,32,33] and incubated for 2 h at 37 °C in 5% CO_2_. After removal of the viral inoculum, cells were maintained in media supplemented with 5% EV-depleted FBS and incubated for 24, 48, 72, 96, and 120 h. Cytopathic effect formation was observed by light field microscopy (Olympus IX71 inverted microscope; Olympus Corp. Miami, FL, USA). Images were taken with a digital camera (Olympus DP72) attached to the microscope and analyzed with ImageJ software version 1.50i (Wayne Rasband, National Institutes of Health, Bethesda, MA, USA). The cell infection was evaluated by ZIKV envelope (E) protein detection as described below.

### 2.5. ZIKV Envelope (E) Protein Detection in ZIKV-Infected Cells by Cytofluorometry (FACS)

The ZIKV E protein detection at the cell membrane’s surface was performed as follows: The different cells were removed from the cell culture plates (C6/36, THP-1, or HMEC-1 were scrapped and homogenized by vigorous pipetting) in sterile 1.5 mL microcentrifuge tubes (Labcon, Petaluma, CA, USA), centrifuged (Eppendorf Centrifuge 5415 R; Eppendorf International, Hamburg, Germany) at 550× *g* for 10 min at 4 °C, and separated from the medium. The cell pellets were fixed with 2% paraformaldehyde (Sigma) for 5 min at 4 °C, blocked for nonspecific binding sites with 2% bovine serum albumin (BSA; Biowest) for 30 min at RT, and washed with 0.5% BSA. The samples were stained with mouse anti-ZIKV E protein clone 1,413,267 antibody (Catalog #CABT-B8528, CD Creative Diagnostics, New York, NY, USA) at a 1:300 dilution in 0.5% BSA and incubated overnight at 4 °C with constant 1000 rpm agitation (Vibrax VXR basic; IKA, Wilmington, NC, USA). After washing with 0.5% BSA and centrifugation, the Alexa Fluor 555-conjugated anti-mouse IgG (H + L), a highly cross-adsorbed secondary antibody (Catalog #A-31570, Thermo Fisher Scientific), was added at a 1:500 dilution in 0.5% BSA, incubated for 2 h at room temperature (RT) with constant 1000 rpm agitation, washed with 0.5% BSA, and centrifugated. The samples were suspended in 300 μL of 0.5% BSA and analyzed by the FACSCalibur flow cytometer (BD Biosciences, San Jose, CA, USA) with CellQuest software. The mock cells were treated in the same way as the infected cells (the mock FACS average values were rested from the infected FACS values).

### 2.6. ZIKV Envelope (E) Protein Detection in ZIKV-Infected Cells by Immunofluorescence (IF)

To confirm the ZIKV E protein presence at the cell membrane surface level, an IF assay was performed as follows: The mosquito C6/36 cells were seeded on an 8-well separation chamber slide system (Lab-Tek II; Thermo Fisher Scientific), incubated until confluence, and infected as described above. The cells were fixed with 2% paraformaldehyde for 5 min at 4 °C, blocked for nonspecific binding sites with 2% BSA for 30 min at RT, and washed with 0.5% BSA. The cells were stained with mouse anti-ZIKV E protein antibody at a 1:300 dilution in 0.5% BSA and incubated overnight at 4 °C with constant 1000 rpm agitation. After washing with 0.5% BSA, the Alexa Fluor 555-conjugated anti-mouse IgG was added at a 1:500 dilution in 0.5% BSA, incubated for 2 h at RT, and washed with 0.5% BSA. The separation chamber was then withdrawn, the slide was covered with mounting medium with DAPI (FluoroQuest; AAT Bioquest, Sunnyvale, CA, USA), and a coverslip (Corning) was placed. The samples were observed via fluorescence microscopy (Olympus IX71 inverted microscope; Olympus Corp.), and the images were analyzed with ImageJ software.

### 2.7. Mosquito C6/36 EVs Isolation from the Cell Culture Medium by Ultracentrifugation

Mosquito C6/36 cells were seeded in cell culture flasks T75 (Corning). The EV isolation from the culture media of the mock and ZIKV-infected cell culture flasks were performed by ultracentrifugation (Appendix A). Briefly, C6/36 cells culture media (50 mL) were collected in sterile conical tubes and centrifuged (GH3.8 rotor, Beckman GPR Centrifuge, Beckman Coulter, Inc.) at 900× *g* for 10 min at 4 °C. The viable cell pellet was discarded. The supernatant was transferred to sterile conical tubes and centrifuged at 2000× *g* for 10 min at 4 °C. The debris pellet was discarded. The supernatant was transferred to 25 × 89 mm centrifuge tubes (Beckman Coulter, Inc.) and centrifuged (SW28 rotor, Beckman XL-90 Centrifuge) at 10,000× *g* for 35 min at 4 °C. The lEVs pellet was suspended in 1 mL of PBS at 4 °C and used immediately or stored at −72 °C (Revco ULT1786, Thermo Fisher Scientific). The supernatant was transferred to centrifuge tubes and centrifuged at 120,000× *g* for 70 min at 4 °C. The supernatant was then discarded. The sEVs and contaminant protein pellet was washed in 5 mL of PBS at 4 °C and incubated for 30 min at RT with constant 100 rpm agitation. The suspension was filtered with a 0.22 μm pore filter, and 25 mL of PBS was added. The samples were transferred to centrifuge tubes and centrifuged at 120,000× *g* for 70 min at 4 °C. The last supernatant obtained in this process was separated and identified as non-EV ZIKV SNT (used as a control in the EV stimulation assays) and stored at −72 °C. The sEVs pellet was suspended in 1 mL of PBS at 4 °C and used immediately or stored at −72 °C [22,34].

### 2.8. Characterization of EVs from Mosquito C6/36 Cells by Nanoparticle Tracking Analysis (NTA)

The characterization of EVs was performed by the detection of reflected light emitted by the Brownian motion of nanoparticles suspended in solution by nanoparticle tracking analysis (NTA) with the help of the NanoSight NS300 equipment and Malvern Instruments software. The optimal detection conditions were previously established (Appendix A), and quantitative controls with 100 and 200 nm polystyrene microspheres (NTA4088 and NTA4089, Malvern Panalytical Products, Mexico City, Mexico) were used (Appendix A). The nanoparticle concentration values (particles/mL) and the size (nm) were determined for each measurement. The ZIKV virions were detected (Appendix A) to identify their presence in the EV samples from ZIKV-infected mosquito cells. In a parallel assay, the nanoparticles present in PBS and in FBS-EV-depleted were quantified to rest the number of nanoparticles obtained in the mock and ZIKV-infected C6/36 EV isolates.

### 2.9. C6/36 lEVs Phosphatidylserine (PS)+ Detection by an Annexin-V Binding Assay

Phosphatidylserine (PS) is located on the cytoplasmic surface of the cell plasmatic membrane, and, during the lEVs biogenesis, PS is translocated from the inner to the outer leaflet of the membrane, exposing the PS that can be detected by the Annexin-V binding assay [35]. The lEVs samples (50 µL) from C6/36 cells were suspended in 200 µL of the Annexin-V binding buffer 1× (Catalog #556454, BD Pharmingen, BD Biosciences, San Jose, CA, USA) that contained fluorescein isothiocyanate (FITC)-conjugated Annexin-V (Catalog #640906, BioLegend, San Diego, CA, USA) at a 1:200 dilution and were incubated for 20 min at RT with constant 100 rpm agitation. After a wash with 250 µL of PBS and centrifugation at 10,000× *g* for 35 min at 4 °C, samples were suspended in 300 µL of PBS. Polystyrene microspheres of a 1 µm diameter size (Polysciences, Inc., Warrington, PA, USA) were used as a FACS calibration control. The samples were analyzed by a FACSCalibur flow cytometer.

### 2.10. Mosquito C6/36 Cell Tetraspanin CD63-Like Protein Detection by FACS

Tetraspanin-like proteins in arthropod cells, including mosquito C6/36 cells, have been described previously [16,30,36,37,38]. Briefly, the C6/36 cells (mock and ZIKV-infected cells) were collected from the cell culture plates in sterile 1.5 mL microcentrifuge tubes, centrifuged at 550× *g* for 10 min at 4 °C, and separated from the medium. The cell pellets were fixed with 2% paraformaldehyde for 5 min at 4 °C, permeabilized with 0.1% Triton X-100 (Sigma) for 5 min at 4 °C, blocked for nonspecific binding sites with 2% BSA for 30 min at RT, and washed with 0.5% BSA. The cells were stained with the phycoerythrin (PE)-conjugated mouse anti-human CD63 antibody (Catalog #557305, BD Pharmingen) at a 1:20 dilution in 0.5% BSA and incubated for 1 h at RT with constant 1000 rpm agitation. The samples were suspended in 300 μL of 0.5% BSA and analyzed by using the FACSCalibur flow cytometer. The mouse IgG1 kappa (P3.6.2.8.1) antibody (Catalog #14-4714-82, eBioscience, San Diego, CA, USA) was used as an isotype control. Isotype FACS average values were rested from the mock and the infected cells FACS values.

### 2.11. Mosquito C6/36 Cells Tetraspanin CD63-Like Protein Detection by Immunofluorescence Assay

To confirm the CD63-like protein presence at the cells membrane’s surface and in the cytosol, the immunofluorescence (IF) assay was performed as follows: The C6/36 cells were seeded on an 8-well separation chamber slide system, incubated until confluence, and infected as described above. The cells were fixed with 2% paraformaldehyde for 5 min at 4 °C, permeabilized with 0.1% Triton X-100 (Sigma) for 5 min at 4 °C, blocked for nonspecific binding sites with 2% BSA for 30 min at RT, and washed with 0.5% BSA. The cells were stained with the PE-conjugated mouse anti-human CD63 antibody at a 1:20 dilution in 0.5% BSA and incubated for 1 h at RT. The mouse IgG1 kappa antibody was used as an isotype control. Finally, the mock and the ZIKV-infected cells were stained with mouse anti-ZIKV E protein antibody at a 1:300 dilution in 0.5% BSA and incubated overnight at 4 °C with constant 1000 rpm agitation. After a wash with 0.5% BSA, the FITC-conjugated anti-mouse IgG (H + L), a highly cross-adsorbed secondary antibody (Catalog #AP308F, Merck, Kennersburg, NJ, USA), was added at a 1:500 dilution in 0.5% BSA, incubated for 2 h at RT with constant 100 rpm agitation, and washed with 0.5% BSA. The separation chamber was withdrawn, the slide was covered with mounting medium with DAPI, and a coverslip was placed. The samples were observed by fluorescence microscopy, and the images were analyzed with ImageJ software.

### 2.12. C6/36 sEVs CD63+ Detection (FACS) by Coupling to Anti-CD63-Coated Paramagnetic Nanobeads

To identify the presence of the tetraspanin CD63 (sEVs marker) on the sEVs membrane’s surface, the C6/36 sEVs isolates were coupled with anti-CD63-coated paramagnetic nanobeads (Catalog #10606D; Invitrogen, Thermo Fisher Scientific) to be detected by cytofluorometry (Appendix A). Briefly, in sterile round-bottom microcentrifuge tubes (Labcon), 100 μL of the sEVs suspension and 20 μL of the paramagnetic nanobeads were added. The samples were incubated for 24 h at 4 °C with constant 1000 rpm agitation. After a wash with 300 μL with PBS, magnetic separation of the bead-coupled sEVs from the matrix suspension was performed using a DynaMag-2 magnet (Life Technologies, Thermo Fisher Scientific), and the supernatant was discarded. The bead-coupled sEVs were suspended in 300 μL of 0.5% BSA. For CD63 detection by FACS, the PE-conjugated mouse anti-human CD63 antibody was used as described above. The samples were analyzed using a FACSCalibur flow cytometer. The paramagnetic nanobeads were treated in the same way as the bead-coupled sEVs (the paramagnetic bead FACS average values of the bead-coupled sEVs were rested from mock and ZIKV-infected cells FACS values).

### 2.13. C6/36 EVs Morphological Characterization by Transmission Electron Microscopy (TEM)

The C6/36 EV isolates were fixed with a 1:1 mixture of 2.5% glutaraldehyde (Electron Microscopy Sciences [EMS], Hatfield, PA, USA) and 4% paraformaldehyde (Sigma) for 2 h at RT and washed three times (5 min each) with PBS. After fixation, the samples were incubated with 2% osmium tetroxide (Alfa Aesar, Thermo Fisher Scientific) for 90 min at RT. The fixed pellets were washed three times with PBS and dehydrated in an ascending graded series of ethanol (30, 50, 70, 80, 90, and 96%), including three passes (5 min each) in absolute ethanol (J.T.Baker) at RT. Three passes (5 min each) in propylene oxide (Sigma-Aldrich) at RT were then performed. The samples were placed in a 1:1 mixture of propylene oxide/epoxy resin for 18 h at RT and embedded in pure epoxy resin (EMS) at 60 °C for 48 h. Ultrathin sections (40–50 nm thick) were obtained in an ultramicrotome (Leica EM UC7, Leica Microsystems, Buffalo Grove, IL, USA) and mounted on copper grids (EMS) covered with formvar (EMS). The sections were contrasted with uranyl acetate (Merck, Kennerworth Fort, NJ, USA) for 30 min and lead citrate (EMS) for 10 min at RT. The preparations were observed with a transmission electron microscope (JEM1010 model; JEOL, Peabody, MA, USA) operating at 80 kV. The images were captured with a CCD300-RC camera (DAGE-MTI, Michigan City, IN, USA) adapted to the microscope and analyzed with ImageJ software.

### 2.14. ZIKV E Protein Detection in lEVs Isolates from ZIKV-Infected C6/36 Cells by FACS

The ZIKV E protein detection at the lEVs membrane’s surface was performed as follows: The lEVs isolates were centrifuged at 10,000× *g* for 35 min at 4 °C, and the supernatant was discarded. The lEVs pellets were fixed with 2% paraformaldehyde for 5 min at 4 °C, blocked for unspecific binding sites with 2% BSA for 30 min at RT, and washed with 0.5% BSA. The lEVs were stained with mouse anti-ZIKV E protein antibody at a 1:300 dilution in 0.5% BSA and incubated overnight at 4 °C with constant 1000 rpm agitation. After washing with 0.5% BSA and centrifugation, the FITC-conjugated anti-mouse IgG secondary antibody was added at a 1:500 dilution in 0.5% BSA, incubated for 2 h at RT with constant 1000 rpm agitation, washed with 0.5% BSA, and centrifugated. The samples were suspended in 300 μL of 0.5% BSA and analyzed by the FACSCalibur flow cytometer with CellQuest software. The lEVs from mock cells were treated in the same way as those from the infected cells (the lEVs mock C6/36 FACS average values were rested of the lEVs ZIKV C6/36 FACS values).

### 2.15. ZIKV E Protein Detection in sEVs Isolates from ZIKV-Infected C6/36 Cells by FACS

The ZIKV E protein detection at the sEVs membrane’s surface was performed as follows: The sEVs isolates were coupled with anti-CD63-coated paramagnetic beads as described above. The bead-coupled sEVs were fixed with 2% paraformaldehyde for 5 min at 4 °C, blocked for nonspecific binding sites with 2% BSA for 30 min at RT, and washed with 0.5% BSA. The bead-coupled sEVs were stained with mouse anti-ZIKV E protein antibody at a 1:300 dilution in 0.5% BSA and incubated overnight at 4 °C with constant 1000 rpm agitation. After washing with 0.5% BSA and recovery by magnetic separation, the Alexa Fluor 555-conjugated anti-mouse IgG secondary antibody was added at a 1:500 dilution in 0.5% BSA, incubated for 2 h at RT with constant 1000 rpm agitation, washed with 0.5% BSA, and recovered by magnetic separation. The samples were suspended in 300 μL of 0.5% BSA and analyzed by the FACSCalibur flow cytometer with CellQuest software. The EVs from mock cells were treated in the same way as those from the infected cells (the sEVs mock C6/36 FACS average values were subtracted from the sEVs ZIKV C6/36 FACS values).

### 2.16. RNA Extraction and Purification

The RNA extraction (from ZIKV stock, C6/36 EVs isolates, or cells [C6/36, THP-1, HMEC-1] pellets) was performed with the centrifugation protocol using the QIAamp RNA Mini kit (Qiagen, Hilden, Germany) according to the manufacturer’s recommendations. Briefly, the samples were first lysed under highly denaturing conditions using the lysis buffer. Next, 70% ethanol was added. Preparations were mixed by pulse/vortexing (Lab-Line Vortex Mixer, Alpha Multiservices, Conroe, TX, USA) for 15 s and incubated for 15 min at RT. The entire volume of the preparations was loaded onto the QIAamp Mini spin columns placed in separation tubes to promote the RNA binding to the columns’ membranes. The tubes were centrifuged (Eppendorf 5415 C Centrifuge) at 6000× *g* for 2 min at RT at every step. The contaminants were washed in two steps using the Absorber Waste 1 (AW1) and AW2 buffers that were added onto the columns. The tubes were centrifuged at 6000× *g* for 2 min at RT at each step. Finally, eluent buffer AVE (RNase-free water with 0.4% sodium azide) was added to the columns, and they were centrifugated at 16,000× *g* for 2 min at RT. The RNA filtrates were collected at 4 °C. The RNA was quantified using a NanoDrop ND1000 Spectrophotometer (Thermo Fisher Scientific) with ND-1000 software version 3.5.2. The RT-PCR protocols were performed immediately, as described below, or samples were stored at −72 °C.

### 2.17. ZIKV Inactivation on Viral Stock Samples and ZIKV-Infected C6/36 EVs Isolates

To inactivate the ZIKV virions, viral stock samples were irradiated at 1200 µJ (× 100) in three consecutive cycles on a UV Stratalinker 1800 (Stratagene, San Diego, CA, USA), and genomic RNA was degraded by RNase A activity assays (Appendix A). Briefly, the total RNA in the samples was quantified, and a concentration of 10 µg/mL of RNAse A (DNase and Protease-free; Thermo Fisher Scientific) was added. The mixtures were then incubated for 1 h, 30 min, and 15 min at 37 °C with 5% CO_2_. A 1:1 proportion of RiboLock RNase Inhibitor (40 U/µL; Thermo Fisher Scientific) was used for 15 min at 37 °C with 5% CO_2_ to inhibit RNase A activity. The RNA degradation pattern was visualized on 2% ethidium bromide-stained (Sigma) 1.2% agarose gel (Invitrogen) using a Typhoon FLA 9500 scanner (GE Healthcare, Chicago, IL, USA) with GE control software version 1.0. The images were analyzed with ImageJ software. The inactivated ZIKV (iZIKV) was evaluated by the lytic plaque assay. The mosquito ZIKV-infected C6/36 EV isolates were irradiated at 1200 µJ (× 100) in three consecutive cycles and treated with RNase A at the best condition of incubation (Appendix A). The samples were used immediately or held at 4 °C. These treatments were identified as EVs (lEVs or sEVs) ZIKV C6/36 (RNase + UV) and were quantified by NTA, the RNA extraction was performed for ZIKV RNA detection by RT-PCR (as describe below), and the lytic plaque assay was performed to evaluate their plaque formation ability in Vero cells (as described above).

### 2.18. ZIKV RNA Detection by Polymerase Chain Reaction with Reverse Transcriptase (RT-PCR) 

The ZIKV RNA detection in a single step by the RT-PCR reaction has been previously described [39,40]. The master mix was prepared according to the specifications given by the OneStep RT-PCR kit (Qiagen). The primers ZIKV FW [5′-GCTGGDGCRGACACHGGRACT-3′] (Mfg. ID 275853243, Integrated DNA Technologies [IDT], San Diego, CA, USA)] and ZIKV RV [5′-RTCYACYGCCATYTGGRCTG-3′] (Mfg. ID 275853246, IDT, USA)] developed by Faye et al. [40] were used. The reaction was performed on a GeneAmp PCR System 2400 (Applied Biosystems, Foster City, CA, USA) with the following conditions: pre-PCR at 50 °C for 40 min and 95 °C for 15 min; 35 cycles at 94 °C for 30 s, 45 °C for 30 s, and 72 °C for 1 min; final elongation at 72 °C for 7 min. The amplified cDNA (amplicon of 364 bp) corresponding to the more specific genome region for the ZIKV E protein, which has no cross reaction with other *Flavivirus* [40], was visualized in 2% ethidium bromide-stained 1.2% agarose gel using a Typhoon FLA 9500 scanner with GE control software. The images were analyzed with ImageJ software.

### 2.19. Quantification of the Total Protein from C6/36 EVs Isolates by Micro BCA Protein Assay

The protein quantification from the C6/36 EVs isolates was performed according to the specification given by the Micro BCA Protein Assay kit (Thermo-Fisher Scientific). The calibration curve standards were performed, in dilutions of 1:2, from a concentrated solution of 2 mg/mL of BSA (Thermo-Fisher Scientific). The blanks, standards, and C6/36 EVs samples (150 µL) were added in triplicate to flat bottom 96-well microplates (Corning) following the addition of 150 µL of the kit work reagent mixture. The plate was covered with sealing tape and incubated at 37 °C for 2 h. The absorbances were measured at 562 nm on the Multiskan Ascent spectrophotometer (Thermo Labsystems) with the Ascent software version 2.6. The average of the 562 nm absorbances of the blank samples was rested from the 562 nm reading of each standard and C6/36 EVs samples. The standard curve was used to determine the protein concentration (mg/mL) of each C6/36 EVs sample.

### 2.20. C6/36 EVs Stimulation Assays on Naïve Vero, C6/36, THP-1, and HMEC-1 Cells

Naïve Vero cells were seeded in 24-well culture plates and incubated until confluence. The cells were inoculated with 400 μL of C6/36 EVs isolates in serial log (10-fold) dilutions (in serum-free medium with dilution factors from 10^−1^ up to 10^−22^) in duplicate and incubated for 2 h. The C6/36 EVs inoculum was removed, and each well was washed with PBS. The monolayers were overlaid with 1 mL of DMEM medium containing 1% carboxymethyl cellulose and 2.5% EV-depleted FBS. The rest of the lytic plaque assay methodology was performed as described above.

Naïve cells (C6/36, THP-1, or HMEC-1) were seeded in 12-well culture plates and incubated until confluence (C6/36 and HMEC-1 cells): For each condition, a strip of 4 plates were used and 2.5 × 10^5^ cells/well were added, to collect 1.0 × 10^6^ cells at the end of the assay (Appendix A). The following conditions were applied: 0.10 mg of protein in 250 µL/well of EV isolates from mock C6/36 cells (lEVs and sEVs, separately), 0.10 mg of protein in 250 µL/well of EV isolates from ZIKV-infected C6/36 cells (lEVs and sEVs, separately), 0.10 mg of protein in 250 µL/well of EV isolates from ZIKV-infected C6/36 cells RNase A + UV-treated (lEVs and sEVs, separately), and 0.10 mg of protein in 250 µL/well of non-EV ZIKV SNT. The mock cells and ZIKV-infected cells (MOI 1) were used as negative and positive controls, respectively. All conditions were added with 250 µL of non-supplemented media and incubated for 2 h; afterward, 1.0 mL of supplemented media with 5% EV-depleted FBS was added and incubated according to the best period of time established in the ZIKV infection assay: 48 h (C6/36 cells), 72 h (HMEC-1), or 96 h (THP-1 cells). The cell cultures for each condition (in a row of 4 plates) were collected (by scrapping and homogenization by vigorous pipetting) in sterile 1.5 mL microcentrifuge tubes (1.0 × 10^6^ cells). For each condition, the ZIKV E protein was detected, and the cytopathic effects were observed via light field microscopy. The monocytes (CD11b, CD14, and CD16) and the endothelial vascular cells (CD142 [Tissue Factor, TF], PAR-1, and CD54 [ICAM-1]) were immunophenotyped. The detection of tumor necrosis factor-alpha (TNF-α) mRNA expression by RT-PCR was performed as well, as described below.

### 2.21. Monocyte and Vascular Endothelial Cell Immunophenotyping

The monocytes and endothelial vascular cells from EV stimulation assays were fixed and blocked for nonspecific binding sites as described above. The human monocytes were immunophenotyped, separately, with the mouse PE-conjugated anti-human CD14 antibody (Catalog #325606, BioLegend), the mouse anti-human CD16 antibody (Catalog #555404, BD Pharmingen), and the mouse PE-conjugated anti-human CD11b antibody (Catalog #301306, BioLegend). Likewise, the ECs were immunophenotyped with the mouse FITC-conjugated anti-human CD142 (TF) antibody (Catalog #13133-MM05-F, Sino Biological, USA), the mouse anti-Protease Activated Receptor (PAR-1) antibody (Catalog #sc-13503, Santa Cruz Biotechnology, USA), and the mouse FITC-conjugated anti-human CD54 (ICAM-1) antibody (Catalog # 35-0549-T025, Tonbo Biosciences, USA). For primary antibodies, the Alexa Fluor 555-conjugated secondary antibody was added at a 1:500 dilution in 0.5% BSA. The samples were analyzed by the FACS Calibur flow cytometer.

### 2.22. Monocytes and Vascular Endothelial Cells TNF-α mRNA Expression by RT-PCR 

The RNA extraction was performed from cells collected after EV stimulation assays for the TNF-α mRNA detection by RT-PCR. The master mix was prepared according to the specifications given by the OneStep RT-PCR kit (Qiagen), as described above. The primers TNF-α FW [5′-ACAAGCCTG-TAGCCCATGTT-3′ (Mfg. ID 110182256, IDT, USA)], TNF-α RV [5′-AAAGTAGACCTGCCC-AGACT-3′ (Mfg. ID 170166847, IDT, USA)], GAPDH housekeeping FW [5′-CCATGTTCGTCATGG-GTGTGAACCA-3′ (Mfg. ID 110179057, IDT, USA)], and GAPDH housekeeping RV [5′-GCCAGT-AGAGGCAGGGATGATGTTC-3′ (Mfg. ID 110179058, IDT, USA)] were used. The reaction was performed on a GeneAmp PCR System 2400 with the following conditions: pre-PCR at 45 °C for 60 min and 95 °C for 15 min; 35 cycles at 94 °C for 30 s, 60 °C for 30 s, and 72 °C for 30 s; final elongation at 72 °C for 10 min. The amplicons [TNF-α: 600 bp and glyceraldehyde-3-phosphate dehydrogenase (GAPDH: 294 bp)] were visualized on 2% ethidium bromide-stained 1.2% agarose gel using a Typhoon FLA 9500 scanner with GE control software. The images were analyzed with ImageJ software.

### 2.23. Endothelial Vascular Cells Permeability Assay

The endothelial vascular barrier integrity after the stimulation with EVs from naïve or ZIKV-infected C6/36 cells was evaluated by a Transwell assay. Briefly, sterile polycarbonate tissue culture-treated Transwell inserts (12 mm) with a 0.4 µm microporous membrane pore size (Corning) were used. The HMEC-1 cells (3.5 × 10^5^) were seeded in the inserts’ upper chambers; meanwhile, the lower chambers were filled with MCDB-131 medium supplemented with 5% EVs-depleted FBS. The cultures were incubated for 48 h at 37 °C with 5% CO_2_ to reach confluence. The EVs samples (0.10 mg of protein in 250 µL) from mock C6/36 and ZIKV C6/36 (RNase + UV-treated or untreated), as described above, were applied in the upper chambers (Appendix A). The inserts were incubated for 2 h; afterward, 100 µL of the medium supplemented with 5% EVs-depleted FBS were added in the upper chambers and incubated for 48 h at 37 °C with 5% CO_2_. The mock HMEC-1 cells, non-EV ZIKV SNT (0.10 mg of protein in 250 µL), and ZIKV-infected HMEC-1 (MOI 1) were used as controls. To determine the cellular permeability degree, FITC (40 kDa)-Dextran (Catalog #60842-46-8, Sigma-Aldrich) was diluted to 1:60 in MCDB-131 medium supplemented with 5% EV-depleted FBS; afterward, 250 µL was added to each upper chamber. An empty insert (without cells) with FITC-Dextran was identified as the no-cell control (NCC), corresponding to 100% of permeability. The inserts were incubated for 1 h at 37 °C with 5% CO_2_. The fluorescence emitted by the FITC-Dextran solution that passes through the cell monolayer to the lower chamber was measured as follows: 100 µL of each lower chamber medium was separated and diluted to 1:50; afterward, the media were passed to a black 96-well plate (Merck) that was analyzed at 492/520 nm in a Synergy H4 hybrid multi-mode microplate reader (BioTek Instruments Inc., USA) with the Gen5 software version 2.09. The permeability percentage (p%) was calculated according to the following formula: p% = (Abs InsertX/Abs NCC) × 100, where *Abs InsertX* corresponds to the absorbance of each different EV stimulus condition or control, and *Abs NCC* corresponds to the absorbance of the no-cell control.

### 2.24. Statistical Analysis

Quantitative data were obtained by three independent assays. The FACS data analysis was performed with FlowJo software version 10.6.1 (BD Biosciences). The bars graphs were obtained by GraphPad Prism software version 8.1.1 (GraphPad Software Inc., USA). Values were expressed as the mean ± standard deviation (SD) and evaluated using an unpaired Student’s t-test with Welch’s correction (for the data to compare two means, assuming unequal SDs). The statistical significance was recognized as *, +, ¡, #, !,/, ~, or ° when *p* < 0.05, **, ++, ¡¡, ##, !!,//, ~~, or °° when *p* < 0.01, and ***, +++, ¡¡¡, ###, !!!,///, ~~~, or °°° when *p* < 0.0001.

## 3. Results

### 3.1. ZIKV Infects C6/36 Mosquito Cells

The ZIKV infection of C6/36 mosquito cells was evaluated by the presence of the ZIKV E protein (FACS) at the cell membrane surface at 24, 48, 72, 96, and 120 h PI. We found that, when using a multiplicity of infection (MOI) of 1 at 48 h PI, the mosquito cells had high levels (38.60 ± 1.05%) of ZIKV E protein at the membrane surface level (Figure 1A).

Likewise, a high percentage (40%) of ZIKV-infected C6/36 cells at 48 h PI was present (Figure 1B), when comparing the E protein levels with the mock cell, and these data were statistically significant (*p* < 0.0001). After 48 h PI, less fluorescent cells were observed in the microscopy assay (Appendix A). It could be that, after main viral infection time, most syncytia structures were in development. The ZIKV-infected C6/36 cultures (48 h PI) developed a more cytopathic effect with the formation of syncytium structures but without the detachment of the cell monolayer, showing increased viral E protein-positive cells determined by fluorescence microscopy (Appendix A). Since the optimal cell infections (activation) were obtained in these conditions (MOI 1, 48 h PI), they were used for all subsequent infection experiments. Recently, it was shown that, during the virus infection process, infected cells are activated and produce different subtypes of EVs, which may have different functions when interacting with other cells, modifying naïve cellular behavior [41]. Therefore, we evaluated whether ZIKV-infected C6/36 mosquito cells could release large and/or small EVs. EVs produced from other arbovirus-infected cells were able to mediate the cell-to-cell communication between vector-host cells [36].

### 3.2. ZIKV-Infected C6/36 Cells Release Large EV Phosphatidylserine+ ZIKV E Protein+

Large EVs were developed by the outward shedding of the cell plasma membrane during cell activation and have a size greater than 200 nm, which can be identified by an Annexin-V binding assay (by FACS). This assay exposes phosphatidylserine (PS) on the outer plasma membrane leaflet [42,43]. The lEVs released from mosquito ZIKV-infected cells and mock cells were isolated from culture supernatants as described above. The characterization of all large EVs was performed by nanoparticle tracking analysis (NTA) by using the NanoSight NS300 equipment and Malvern Instruments software. The different experimental conditions for EVs detection were first established (Appendix A), and quantitative controls with 100 and 200 nm polystyrene microspheres (NTA4088 and NTA4089, Malvern) were used (Appendix A). Nanoparticles present in PBS and EVs-depleted FBS were also quantified to correct the number of the isolated EVs. The ZIKV virions were detected (as a peak of size of 63.5 ± 8.1 nm) to identify its presence in the EVs isolates from ZIKV-infected mosquito cells (Appendix A).

For the NanoSight 300, a camera level of 12.0 was used with a detection limit of 2.5, a temperature of 20 °C, samples diluted in 1.0 mL of PBS, reading periods of 30 s, and three consecutive repetitions. The NTA from lEVs isolates of ZIKV-infected cells showed a concentration of 2.92 × 10^10^ ± 3.55 × 10^9^ particles/mL with an average value of size of 319.3 ± 11.5 nm. Compared with NTA from lEVs isolates of mock cells (2.14 × 10^9^ ± 4.10 × 10^8^ particles/mL with an average value of size of 268.9 ± 8.2 nm), the concentration of nanoparticles from lEVs isolates from infected cells were 13.6-fold higher (Figure 2A).

In parallel, we observed that, during ZIKV infection (MOI 1, 72h PI) of the C6/36 cells (Figure 2B), the percentage of lEVs PS+ released to the cell culture supernatant (60.50 ± 0.87%) was 1.98-fold higher (*p* < 0.001) than lEVs PS+ from uninfected (30.60 ± 0.96%) mosquito cells (Figure 2C). The fluorescence emitted by Annexin-V binding was proportional to the PS presence: The ZIKV-infected C6/36 lEVs mean fluorescence intensity (MFI) was compared with that of the microbead control and mock C6/36 lEVs values (Appendix A). The Annexin-V binding MFI in the lEVs from ZIKV-infected cultures was 1.45-fold higher than that of the lEVs from the mock cultures.

The lEVs samples were also characterized by transmission electron microscopy (TEM): The analysis of images from the mock C6/36 lEVs and ZIKV-infected C6/36 lEVs (Figure 2D) showed different EV populations, which were heterogeneous in shape, with sizes up to 200 nm (1000 nm scale) and a defined membrane in a proper structural resolution. These data agree with other reports, which demonstrated that cells secrete EVs as a heterogeneous population with different sizes and shapes [30,44]. We did not identify viral particles inside lEVs TEMs from ZIKV-infected cells, so we evaluated the ZIKV E protein presence on the membrane’s surface of lEVs. We found 18.27 ± 1.27% of positive ZIKV E protein+ lEVs (*p* < 0.01) compared with the lEVs from the mock C6/36 cells (Figure 2E,F). The ZIKV E protein (MFI) in the lEVs from ZIKV-infected cells was 18.6-fold higher than that of the lEVs from the mock cultures (Appendix A).

### 3.3. ZIKV-Infected C6/36 Cells Release Small EV CD63-Like+ ZIKV E Protein+

First, the presence of CD63-like tetraspanin at the membrane surface of C6/36 cells was evaluated as well. The point time of the CD63 membrane decreased (CD63 internalization) to select the best condition for the identification of the sEVs marker (inside cells) and, likewise, the time for the optimal isolation of sEVs CD63-like+ [45]. The tetraspanin detection (FACS) was evaluated at 24, 48, 72, 96, and 120 h PI (Figure 3A). We observed that naïve C6/36 cells constitutively contain high levels of the CD63-like tetraspanin (Figure 3A,B) at the cell membrane surface (55.67 ± 2.30%). Nevertheless, in ZIKV-infected cultures, an increase in the CD63 percentage (79.40 ± 2.78%) was present at 24 h PI (1.4-fold higher), while the highest percentage (88.73 ± 1.35%) of the CD63+ cells were obtained at 48 h PI (1.6-fold higher compared with mock cells). The increased tetraspanin values were significant (*p* < 0.0001). We found that CD63 levels were decreased at 72–96 h PI (Figure 3A,B). These data suggest that CD63-tetraspanin internalization may occur after 48 h PI. Therefore, the sEVs biogenesis in mosquito cells could take place between 48 and 72 h. The sEVs CD63+ were then isolated at 72 h PI. The presence of the CD63 tetraspanin was also evaluated (red) by fluorescence microscopy (100×) in mock and ZIKV-infected C6/36 cultures (green for ZIKV E protein). The presence of the CD63 tetraspanin inside C6/36 cells suggests the endosomal nature of small EVs (Appendix A).

The NTA from sEVs isolates of ZIKV-infected cells showed a concentration of 3.17 × 10^11^ ± 5.62 × 10^10^ particles/mL with an average value of size of 125.5 ± 1.6 nm. Compared with the NTA from sEVs isolates of mock cells (2.39 × 10^10^ ± 4.41 × 10^9^ particles/mL with an average value of size of 107.8 ± 3.1 nm), the concentration of nanoparticles from sEVs isolates from infected cells were 13.3-fold higher (Figure 3C).

The sEVs isolates were then identified by positive selection, using paramagnetic nanobeads coated with anti-CD63 antibodies (Appendix A). Previously, we determined whether ZIKV viral particles could cross-react with the paramagnetic nanobeads (Appendix A). We found that ZIKV did not couple to the nanobeads, so the sEVs CD63+ detection is free of viral particles. The sEVs CD63+ percentage from ZIKV-infected cultures (19.32 ± 0.93%) was 1.7-fold higher than the sEVs CD63+ from the mock cultures (11.08 ± 0.34%), showing a significant difference (*p* < 0.01) (Figure 3D,E). The MFI values (obtained by FACS) were proportional to the sEVs CD63+ presence (Appendix A). We found that the sEVs CD63+ from ZIKV-infected cells showed MFI values 1.3-fold higher than the sEVs CD63+ from the mock cultures. 

The TEM images (Figure 3F) from the mock and ZIKV-infected C6/36 sEVs (500 nm scale) show a heterogeneous population of sEVs [46] in terms of their size (fewer than 200 nm in diameter), shape, and content; this population is also well defined by a bilipid membrane. We did not identify viral particles inside sEVs TEMs from ZIKV-infected cells, so we evaluated the ZIKV E protein presence on the membrane’s surface of sEVs. We found 31.19 ± 0.28% of positive ZIKV E protein+ sEVs coupled with paramagnetic nanobeads (*p* < 0.001) compared with the sEVs from the mock C6/36 cells (Figure 3G,H). The ZIKV E protein (MFI) in the sEVs from ZIKV-infected cells was 4.03-fold higher than that of the sEVs from the mock cultures (Appendix A).

### 3.4. ZIKV C6/36 EVs, after ZIKV Inactivation, Carry Viral RNA, Reproduce Lytic Plaque Formation on Vero Cells, and Favor Infection in Naïve Mosquito Cells

We determined whether ZIKV mosquito EVs participate during the infectious process as was recently reported for other arthropod-borne flaviviruses [16,30,36]. It was then determined whether ZIKV C6/36 EVs contain viral elements as RNA or E protein and whether they support the infection of naïve cells.

First, we determined whether ZIKV could be inactivated by RNAse A activity and UV radiation at 1200 µJ (× 100) in three consecutive cycles (Appendix A). We evaluated different conditions for incubation times and found that the complete RNA degradation from ZIKV viral stock occurred for 1 h at 37 °C (Figure 4A). The infection capability of inactivated ZIKV (iZIKV) was evaluated by a lytic plaque assay (Figure 4B) and found no lytic plaque formation after the inactivation process.

We then proceeded to inactivate free ZIKV virions in C6/36 mosquito EV isolates (small/large). These samples were irradiated at 1200 µJ (×100) in three consecutive cycles by using an UV Stratalinker (Appendix A). In addition, the possible presence of ZIKV genomic RNA in the EVs samples, as a possible precipitation product during the EVs isolation process, was eliminated with RNase A, as described above: By using 10 µg/mL of RNase A (DNase and Protease-free) added to the different EV isolates for 1 h at 37 °C, we observed total RNA degradation in all samples (sEVs and lEVs) (Figure 4A).

To evaluate the integrity of the EVs (sEVs and lEVs) after the ZIKV inactivation process, we proceed to quantified them by NTA. The NTA from lEVs ZIKV C6/36 (RNase A + UV) showed a concentration of 2.48 × 10^10^ ± 4.07 × 10^9^ particles/mL with an average size value of 304.1 ± 10.9 nm, while the NTA from sEVs ZIKV C6/36 (RNase A + UV) showed a concentration of 2.49 × 10^11^ ± 2.29 × 10^10^ particles/mL with an average size value of 150.9 ± 5.5 nm (Figure 4C). The NTA histograms showed the same patterns of the NTA histograms shown in Figure 2A and Figure 3C, so the EVs’ integrity is preserved. In Figure 4C, the peak containing particles of nearly 50 nm, compatible with ZIKV, was substantially reduced.

Next, to evaluate the possible ZIKV genomic RNA presence inside small and large EVs (RNase A + UV-treated), all samples were processed for RNA extraction and purification by using the QIAamp RNA Mini kit, according to the manufacturer’s instructions. The samples were first lysed under highly denaturing conditions using the lysis buffer. The ZIKV-RNA amplification was performed by RT-PCR, according to the specifications given by the OneStep RT-PCR kit (see Materials and Methods). The amplified cDNA corresponded to the specific 364 bp E-amplicon for the ZIKV [40] envelope protein, which was visualized on 2% ethidium bromide-stained 1.2% agarose gel (Figure 4D). These findings suggest that both small and large EVs from ZIKV-infected mosquito cells may carry viral RNA after RNase A + UV treatment.

As a result of the significance of these data, we also evaluated the possible mammalian naïve cell infection via small/large ZIKV C6/36 EVs. With this aim, first, we used the epithelial cells (Vero) from the monkey *Cercopithecus aethiops* (used as gold standard cells for infection assay) [32] to perform plaque assays (as described above) in the presence of small and large ZIKV C6/36 EVs (also RNase A + UV-treated and untreated samples). The presence of lytic plaques in ZIKV-infected Vero cells was present in high or undetermined amounts at different dilutions (Figure 4E). Importantly, lytic plaques were also observed in cultures in the presence of small and large ZIKV C6/36 EVs isolates in a more concentrated amount, which were also formed in high or undetermined quantities. However, plaque formation was not detected in the negative control named non-EVs ZIKV SNT (the final supernatant obtained in the last centrifugation during the isolation of EVs from ZIKV-infected C6/36 cell culture media). These data suggest the ZIKV infection of monkey epithelial cells via lEVs/sEVs released from ZIKV-infected C636 mosquito cells.

Therefore, different EVs stimulation assays were performed using naïve C6/36 cells in the presence of ZIKV (MOI 1), mock C6/36 EVs, or ZIKV C6/36 small/large EVs (including RNase A + UV-treated and untreated isolates) (Appendix A). The following conditions were applied: 0.10 mg of protein in 250 µL/well of EVs isolates from mock C6/36 cells (lEVs and sEVs, separately), 0.10 mg of protein in 250 µL/well of EVs isolates from ZIKV-infected C6/36 cells (lEVs and sEVs, separately), 0.10 mg of protein in 250 µL/well of EVs isolates from ZIKV-infected C6/36 cells RNase A + UV-treated (lEVs and sEVs, separately), and 0.10 mg of protein in 250 µL/well of non-EVs ZIKV SNT. The mock cells and ZIKV-infected cells (MOI 1) were used as negative and positive controls, respectively. We also evaluated the iZIKV infection capability on C6/36 cells (Appendix A), and we found that iZIKV did not infect naïve C6/36 cells after 48 h of incubation, because the ZIKV E protein was undetectable.

The samples were incubated over 48 h (the best infection time) and evaluated for possible viral infection by means of ZIKV E protein detection using a FACS assay. The ZIKV E protein presence was detected in 44.39 ± 0.69% of ZIKV-infected C6/36 cells (Figure 5A–C), showing statistical significance (*p* < 0.0001) in relation to the mock cells (*) (34.7-fold higher). In naïve C6/36 cells stimulated with ZIKV C6/36 lEVs, the E protein was found in 38.24 ± 1.15% of cells and 28.44 ± 1.37% of cells stimulated with ZIKV C6/36 lEVs (RNase A + UV-untreated and treated) (#,!), showing statistical significance (*p* < 0.0001) compared with the mock C6/36 cells and stimuli in the presence of mock C6/36 EVs (Figure 5A,B). Similarly, in naïve mosquito cells cultured with ZIKV C6/36 sEVs, the viral E protein was found in 39.31 ± 0.58% of cells and 33.00 ± 0.29% of cells stimulated with ZIKV C6/36 sEVs (RNase A + UV-untreated and treated) (/, ~), showing statistical significance (*p* < 0.0001) compared with the mock cells and naïve cells stimulated with EVs from mock C6/36 cells (Figure 5A–C). The present data suggest that ZIKV C6/36 (large and small) EVs could favor the infection of mosquito naïve cells.

Likewise, in parallel assays, all samples were evaluated for the presence of ZIKV E protein by fluorescence microscopy and by the cytopathic effects observation using light field microscopy (Appendix A). The cytopathic effects in the light fields were indicated with black arrows (20×). An increased cytopathic effect with the formation of syncytium structures was present in the ZIKV-infected cultures but also in cultures stimulated with small and large ZIKV C6/36 EVs (RNase A + UV-treated and untreated) isolates. The ZIKV E protein (red) was detected in naïve C6/36 cells stimulated with the small and large ZIKV C6/36 EVs at similar levels and patterns of ZIKV-infected mosquito cells (fluorescence microscopy, 60×) and were undetected in cultures of naïve C6/36 cells in the presence of mock C6/36 EVs (small and large). The same was found for the non-EVs ZIKV supernatant or mock C6/36 cultures.

### 3.5. ZIKV-Infected C6/36 EVs Participate during Infection of Naïve Human Monocytes

Recently, it was reported that dengue virus (DENV) uses small EVs of C6/36 mosquito cells for its transmission from the vector to mammalian host cells, including human skin keratinocytes and ECs [16,30]. However, to date, it has not been determined if EVs from ZIKV-infected C6/36 cells participate during monocyte infection. As monocytes are the main target cells during ZIKV human host infection [17,47], we evaluated the possible participation of large and small EVs in the potential. Initially, we evaluated the viral infection of human monocytes (MOI 1) via the detection of the viral E protein at the surfaces of the membrane monocytes at 24, 48, 72, 96, and 120 h PI using a FACS assay (Figure 6).

ZIKV was able to establish a productive infection of human monocytes, since the viral E protein was detected at high levels on the cell membrane’s surface in all post-infection time points of the assay (Figure 6A). However, the greater percentages of ZIKV-positive cells were present at 24 h PI (79.98 ± 1.07%), at 96 h PI (77.92 ± 1.17%), and at 120 h (78.01 ± 0.98%), which were statistically significant (*p* < 0.0001) when compared to the mock cells. Likewise, we found that the ZIKV infection of THP-1 human monocytes favor progressive activation and cell differentiation effects (Appendix A). The cytopathic effect observation by light field microscopy supports the presence of higher amounts of adherent cells between 96 and 120 h PI. Consequently, we decided to evaluate the ZIKV infection of naive monocytes after the EVs stimulation at 96 h PI.

Different EVs stimulation assays were performed using naïve THP-1 cells in the presence of ZIKV (MOI 1), mock C6/36 EVs, or ZIKV C6/36 EVs (small and large isolates as the same for the RNase A + UV-treated and untreated samples) (Appendix A), which were evaluated for possible infection by means of ZIKV E protein detection by FACS (see Materials and Methods). We also evaluated the iZIKV infection capability on THP-1 cells (Appendix A) and found that iZIKV did not infect naïve monocytes after 96 h of incubation, because the ZIKV E protein was undetectable.

As was expected, the viral E protein was detected in 73.41 ± 0.59% of ZIKV-infected monocytes and was statistically significant (*p* < 0.0001) when compared to the mock cells (*) (Figure 7A–C). The ZIKV E protein was also detected in higher amounts on naïve THP-1 cells that were stimulated with ZIKV C6/36 lEVs (54.89 ± 0.56% in RNase A + UV-treated (!) isolates and 70.23 ± 1.53% in untreated (#) isolates) and ZIKV C6/36 sEVs (38.29 ± 0.79% in RNase A + UV-treated (~) isolates and 41.34 ± 0.39% in untreated (/) isolates). The ZIKV E protein levels were statistically significant (*p* < 0.0001) when compared to the ZIKV-C6/36 EVs stimulated cultures against the mock cells, mock C6/36 EVs, and non-EV ZIKV SNT (°) cultures (Figure 7A–C).

The present data suggest that ZIKV-infected C6/36 EVs not only support the infection of naïve mosquito cells but also participate during infection of mammalian host cells, as in the case of human monocytes, which are important immune effector cells during host–pathogen interplay.

### 3.6. ZIKV-Infected C6/36 EVs Promote Change in Monocyte Phenotype (CD14, CD16, and CD11b)

It is known that EVs may have different functions when interacting with other cells, modifying their naïve cellular behavior [48]. Therefore, if EVs released by ZIKV-infected mosquito cells were able to infect human monocytes, they could also favor monocyte activation and/or differentiation. To assess this objective, naïve human monocytes were stimulated by the presence of ZIKV (MOI 1), mock C6/36 EVs, or ZIKV C6/36 EVs (small and large isolates as the same for the RNase A + UV), which were evaluated for monocyte activation or differentiation (to adherent phenotype cells) by means of the monocytes’ phenotypic shift from a classical (CD14++ CD16−) to an intermediate (CD14++ CD16+) or non-classical (CD14+ CD16++) phenotype, which seems to be the main producer of inflammatory mediators in response to viral infection [49].

By the stimulation assays in the presence of the different EVs samples, we observed (Figure 8A,B) that the classical monocyte phenotype changes to CD14++ CD16+ intermediate monocytes with respect to the mock THP-1 cells and those stimulated with the mock C6/36 EVs (*p* < 0.0001 for CD14 and *p* < 0.01 for CD16). Moreover, in a parallel assay, we observed that naïve monocytes were differentiated and expressed CD11b+ at the membrane’s surface (Figure 8C) with elevated levels in ZIKV-infected monocytes and those stimulated with ZIKV C6/36 EVs (*p* < 0.0001) compared with the mock THP-1 cells and those stimulated with mock C6/36 EVs. On the other hand, naïve monocyte activation by the ZIKV C6/36 EVs was also observed by light-field microscopy (Appendix A), showing the transformation of naïve cells to the adherent phenotype (black arrows), as similar levels occur in ZIKV-infected monocytes.

It has been suggested that monocyte subsets (intermediate and non-classical) play an essential role in immunopathology during *Flavivirus* infection [49], since non-classical monocytes seem to be the main producers of pro-inflammatory mediators in response to viral infection. The present data show that ZIKV C6/36 (small/large) EVs activate and differentiate naïve monocytes, changing cells to a pro-inflammatory state, in a similar mode to ZIKV (MOI 1) infection.

Next, to evaluate the possible expression of a pro-inflammatory response induced by EVs released from ZIKV-infected C6/36 cells, we performed a detection of the tumor necrosis factor-alpha (TNF-α) mRNA expression in naïve monocytes infected by ZIKV (MOI 1), and those stimulated with the mock C6/36 EVs or ZIKV C6/36 EVs (RNase A + UV-treated and untreated isolates). We found that, like ZIKV-infected monocytes, ZIKV C6/36 sEVs were able to induce TNF-α mRNA expression in human monocytes (Figure 8D). TNF-α is a pro-inflammatory cytokine that was recently determined to be an important host factor involved in neurological disorders and central nervous system inflammation during ZIKV human infection [50].

A growing number of evidence indicates that sEVs are involved in inflammatory processes or immune responses that play an important role in a large number of pathologic states, including infectious diseases. sEVs can modulate gene expression and the functions of the cells with which they interact, and their content depends on the cells from which they are released [23,27]. We found that sEVs from ZIKV-infected C6/36 cells induce immunophenotype changing (to intermediate/non-classical) in monocytes. This proinflammatory phenotype could be directly implicated in TNF-α mRNA expression.

### 3.7. ZIKV C6/36 EVs Participate during Infection of Naïve Endothelial Vascular Cells

The ZIKV mosquito EVs participation during infection of vascular ECs is unknown. We first determined whether ZIKV (MOI 1) was able to infect human endothelial vascular cells, by means of the viral E protein detection on cells membrane surface at 24, 48, 72, 96, and 120 h PI by the FACS assay. First, the optimal conditions for ZIKV vascular endothelial cell infection were evaluated (Figure 9).

ZIKV was able to establish a productive infection in microvascular endothelial (HMEC-1) cells (Figure 9A,B), since viral E protein was detected at high levels on the cell membrane surface mainly at 24 (45.62 ± 1.76%) and 72 h (38.22 ± 1.20%) PI time points, with a significance level of *p* < 0.0001 when compared to the mock HMEC-1 cells (22.9- and 19.2-fold higher, respectively). Likewise, when these samples were observed by light-field microscopy, the ZIKV-infected ECs showed important cytopathic effects (black arrows), with a formation of syncytium structures at 48 and 72 h PI, but without monolayer detachment (Appendix A). We also evaluated the iZIKV infection capability on HMEC-1 cells (Appendix A) and found that iZIKV did not infect naïve ECs after 72 h of incubation, because the ZIKV E protein was undetectable.

Afterward, the possible participation of small/large C6/36 EVs in the potential infection of vascular ECs was evaluated. Next, different stimulation assays were performed using HMEC-1 naïve cells in the presence of ZIKV (MOI 1), the mock C6/36 EVs, and the ZIKV small/large C6/36 EVs (the same for the RNase A + UV-treated and untreated). All stimuli were evaluated by measuring the ZIKV E protein presence by FACS (Figure 10).

As shown in Figure 10A, viral E protein was present at high levels in ZIKV (MOI 1) infected HMEC-1 cells (30.22 ± 0.58%), but also at high levels in naïve ECs stimulated by the ZIKV C6/36 sEVs (RNase A + UV-treated (19.28 ± 0.80%) and untreated (26.40 ± 0.78%)) and ZIKV C6/36 lEVs (RNase A + UV-treated (15.28 ± 0.49%) and untreated (26.32 ± 0.56%)). The percentage of viral E protein was compared between all conditions’ values against the mock HMEC-1; these values were statistically significant (*p* < 0.0001) (Figure 10B,C). These data suggest that ZIKV infected-C6/36 EVs (large and small) support the infection of mammalian host cells, including vascular ECs.

### 3.8. ZIKV C6/36 EVs Favor a Pro-Inflammatory and Pro-Coagulant State of Vascular Endothelial Cells and Promote the Endothelial Vascular Cells’ Permeability

Recent observations in humans and animal models [51,52] suggest that, in severe Zika cases, different coagulation disorders occur. It has been shown that some viruses activate the coagulation system through tissue factor (TF) receptor expression [53]. We previously reported that the DENV upregulates the TF coagulation receptor in endothelial vascular cells, which triggers the generation of hemostatic proteases (thrombin) favoring the activation of protease-activated receptors or PARs, which, in turn, induce signaling inflammatory pathways (via phosphorylation of MAPKs p38 and ERK1/2, by transcription of NF-κB factor), thereby supporting the upregulation of VCAM-1 adhesion or pro-inflammatory molecules in ECs [54]. Next, we determined whether the ZIKV infection (MOI 1) of vascular endothelial cells (HMEC-1) or the stimulation by ZIKV C6/36 EVs favor a pro-inflammatory/pro-coagulant state of naïve endothelial vascular cells.

Therefore, we assessed the possible participation of small and large EVs issued from ZIKV-infected C6/36 cells for the induction of coagulation (TF) or inflammation (PAR-1) receptors at the membrane’s surface of ECs. In a parallel assay, the adhesion ICAM-1 molecule was also evaluated. Different stimulation assays were performed using naïve HMEC-1 cells in the presence of ZIKV (MOI 1), the mock C6/36 EVs, or the ZIKV C6/36 EVs (small and large isolates, the same for the RNase A + UV), which were evaluated for the presence of TF, PAR-1, and ICAM-1 at their cell membranes (FACS) at 72 h post-stimulus (Figure 11A–C). Likewise, the cytopathic effect formation on these samples was evaluated by light-field microscopy (Appendix A).

Elevated levels of TF (20.27 ± 0.51%) were detected in ZIKV-infected (MOI 1) vascular ECs (Figure 11A), but also in the presence of large ZIKV C6/36 EVs (16.75 ± 0.59%), which were both statistically significant (*p* < 0.0001) when compared with the mock HMEC-1 cultures (*). In small EV culture samples, the TF values were 4.82 ± 0.23%. The upregulation of the TF receptor may trigger the generation of hemostatic proteases (thrombin) favoring the activation of protease-activated receptors (PARs). Figure 11B shows the activation percentages of the PAR-1 in ECs infected with ZIKV (26.41 ± 0.56%), and the same is true for large ZIKV C6/36 EVs (23.17 ± 0.33% for untreated and 13.15 ± 0.38% for RNase A + UV-treated) and small ZIKV C6/36 EVs (21.55 ± 0.14% for untreated and 16.19 ± 0.79% for RNase A + UV-treated) culture samples. We found a significant difference (*p* < 0.0001) in all EVs stimuli compared with the mock HMEC-1 cultures. It is well known that PAR-1 favors signaling pathways for the expression of pro-adherent and pro-inflammatory molecules [55]. Therefore, we assessed ICAM-1 (Figure 11C) detection on EC surfaces in ZIKV-infected HMEC-1 cultures (12.87 ± 0.16%). The same was observed in cultures of naïve ECs in the presence of the mock C6/36 EVs or the ZIKV C6/36 EVs (small/large isolates, the same for the RNase A + UV) (*p* < 0.001). These data were corroborated by light-field microscopy (Appendix A), where ZIKV-infected ECs showed an increased cytopathic effect (black arrows) with the formation of vacuolization and syncytial structures. Cytopathic effects were also observed in all naïve EC culture assays stimulated by the ZIKV C6/36 EVs.

Moreover, to evaluate the possible expression of the pro-inflammatory response by the stimulation of EVs released from ZIKV-infected C6/36 cells, we measured the TNF-α mRNA expression in naïve ECs infected by ZIKV (MOI 1) and those stimulated by the mock C6/36 EVs and the ZIKV C6/36 EVs (small/large isolates as the same for the RNase A + UV). We found that, similar to ZIKV infection, ZIKV C6/36 EVs were able to induce TNF-α mRNA expression in endothelial vascular cells (Figure 11D). Our data suggest the possible participation of the coagulation–inflammation process in the coagulation disorders present in severe cases of Zika. The endothelial vascular cell activation (damage) during ZIKV infection with an inflammatory response can cause EC dysfunction and weaken the endothelial barrier integrity. Thus, we evaluate the vascular endothelial barrier integrity in vitro using a Transwell assay (Figure 11E).

Our data indicate that endothelial vascular cells are susceptible to ZIKV infection and activation by both ZIKV (MOI 1) and ZIKV (small/large) C6/36 EVs with pro-inflammatory cytokine expression, which could increase endothelial monolayer permeability. Therefore, we determined whether ZIKV infection or ZIKV C6/36 EVs disturb the vascular endothelial barrier’s integrity in vitro using a Transwell assay (see Materials and Methods; Appendix A), performed in a naïve EC culture in the presence of ZIKV (MOI 1), the mock C6/36 EVs, or the ZIKV C6/36 EVs.

For ZIKV-infected ECs, the permeability percentage (15.77 ± 1.23%) increased 2.3-fold compared to the mock HMEC-1 cultures (*p* < 0.0001). In the presence of the ZIKV C6/36 lEVs (RNase A + UV-treated (10.87 ± 1.29%) and untreated (10.95 ± 1.37%)), permeability increased, on average, 1.6-fold (*p* < 0.05); in cultures stimulated by ZIKV C6/36 sEVs (RNase A + UV-treated (10.29 ± 1.14%); when untreated (12.33 ± 1.19%)), the endothelial permeability increased 1.5- (*p* < 0.05) and 1.8-fold (*p* < 0.01), respectively (Figure 11E). These data suggest that ZIKV C6/36 EVs may participate in vascular endothelial damage with a weakening of the endothelial barrier integrity and support the mosquito EVs participation during the infection process, which could contribute to the pathogenesis of ZIKV infection in a human host.

## 4. Discussion

Vector-borne diseases cause nearly one million deaths per year and represent 17% of all infectious illnesses worldwide [56]. This public health problem highlights the importance of understanding how arthropod vectors, microbes, and their mammalian hosts interact. At present, research efforts are focused on pathogen–host interactions, with a lack of attention on the significant contribution of vector-derived products in disease development. The molecular and cellular events occurring in vector–pathogen–host interactions are critical in determining the outcome of the vector-borne diseases. Recently, it was proposed that one strategy used by vectors, to promote a successful host infection, is the manipulation of EVs [41,57]. Infected vector cells secrete vesicles that may contain antigens, nucleic acids, and microbial cargos (or the whole pathogen), which exacerbate the pathogenesis and modulate the host responses [36]. EVs are involved in the exchange of bioactive molecules between cells. Although all EVs are vesicles constituted by lipid layers, their cargo reflects the state of the source cell, and their content can be altered in adverse conditions or manipulated by pathogens. Extracellular vesicles have an important role in the establishment of arboviral diseases [16,30,36]. EVs originating from arthropod vectors are an important strategy for immune evasion during viral transmission. For example, dengue virus (DENV) uses EVs derived from mosquitoes to infect mammalian cells. Mosquito-derived vesicles carry DENV proteins and a full-length viral genome. DENV transmission may occur through the interaction between the tetraspanin Tsp29Fb, a mosquito homolog of the human sEVs marker CD63, and the viral E protein [16]. Similarly, during the infection of the tick-cell line ISE6 with the Langat virus (LGTV), cells release sEVs that contain cargo from both the virus and the vector, which enable these EVs to transmit the virus to mammalian cells [36]. Ample evidence has been provided to show that sEVs carry and deliver viral genomes into recipient cells in vitro, as was reported for the hepatitis C virus (HCV), the hepatitis A virus (HAV), and human herpes virus 6 (HHV-6), among others [58,59,60,61,62]. Nevertheless, to date, it is unknown if, during mosquito ZIKV delivery to the vertebrate host cells, this arbovirus or the viral components such as viral proteins and the viral genome can be transferred by EVs from infected mosquito tissues to the host. We determined whether viral elements (viral RNA and envelope protein) could be transported by mosquito EVs, as was recently reported for other arthropod-borne flaviviruses.

In the present work, we found that, during ZIKV infection of C6/36 mosquito cells, small and large EVs were produced in high amounts. The isolated sEVs from ZIKV-infected cells were purified using paramagnetic nanobeads coated with anti-CD63 antibodies, thus demonstrating their endosomal origin. The separation of sEVs from virions by positive selection using magnetic beads coupled with an antibody against a tetraspanin enriched in sEVs is by far the best method [60,63]. The EVs characterization by nanoparticle tracking analysis (NTA) and transmission electron microscopy (TEM) from the mock C6/36 and the ZIKV-infected C6/36 EVs (Figure 2D and Figure 3F) showed different EVs populations, which were heterogeneous in shape, with sizes up to 200 nm (1000 nm scale) in the case of lEVs, and less than 200 nm (500 nm scale) in diameter for sEVs. These populations showed well-defined bilipid membranes in a proper structural resolution. Our data agree with other reports demonstrating that cells secrete EVs as a heterogeneous population with different sizes and shapes [30,44,46]. Recently, it was proposed that the EVs size variation could be due to the internal content (proteins/RNAs) of vesicles produced from uninfected and infected cells [30]. By the NTA, we found that the lEVs from ZIKV-infected C6/36 presented an average size of 319.3 ± 11.5 nm (1.2-fold higher compared with the average size of lEVs from the mock cells) (Figure 2A). On the other hand, the sEVs from ZIKV-infected C6/36 presented an average size of 125.5 ± 1.6 nm (1.2-fold higher compared with the average size of sEVs from the mock cells) (Figure 3C). In this sense, the ZIKV C6/36 EVs RNase A + UV treatment assays (Figure 4 and Appendix A), performed to eliminate the possible presence of polluting free virus in samples, suggest a presence of viral RNA inside ZIKV-infected C6/36 EVs. The presence of the ZIKV E protein detected in ZIKV C6/36 lEVs and sEVs isolates was not observed in mock C6/36 EVs (Figure 2E,F and Figure 3G,H). These findings suggest that both sEVs and lEVs from ZIKV-infected mosquito cells carry the viral E protein and viral RNA. Different viruses manipulate EVs for their benefit in order to increase their persistence, pathogenesis, and transmission [58,59,60,61,62,64]. The hijacking of mosquito cell membranes by ZIKV could facilitate their escape from host immune responses, promoting the viral elements’ spread. We first used the epithelial cells (Vero) from the monkey *Cercopithecus aethiops* (used as gold standard cells for infection assay) [32]. The presence of lytic plaques (Figure 4E) in cultures of Vero with the ZIKV C6/36 EVs (lEVs and sEVs) isolates were observed in more concentrated amounts, formed in high or undetermined quantities. However, plaque formation was not detected in the negative controls. These data suggest ZIKV-mosquito EVs participation during the infectious process of monkey epithelial cells. Additionally, the possibility of EVs participation during the infection process of naïve mosquito cells via cell-to-cell EVs transfer was evaluated. We demonstrated the presence of the ZIKV envelope protein at the mosquito cell membrane surface in cultures incubated with the ZIKV C6/36 EVs (large and small), implying the infection of naïve mosquito cells (Figure 5 and Appendix A). These results are consistent with those of Vora et al. (2018), showing that the full-length genome of DENV-2 detected in the EVs from DENV-infected mosquito cells was infectious in naïve mosquito and mammalian cells [16].

To date, it has not been determined whether EVs from ZIKV-infected mosquito cells are utilized as ZIKV viral element vehicles (genome/protein) to mammalian host cells. As monocytes and vascular ECs are the main targets during ZIKV human host infection, we evaluated the possible participation of EVs derived from the ZIKV-infected mosquito C6/36 cells in potential viral element carriers to human monocytes and vascular ECs. ZIKV was able to establish a productive infection in human monocytes and ECs, since the viral E protein was detected at high levels on the cell membrane surface at all PI time points of the assay (Figure 6 and Figure 9). In stimulation assays of monocytes and ECs in the presence of ZIKV-infected C6/36 EVs, we found that ZIKV C6/36 EVs supported cell infection in naïve cells (Figure 7 and Figure 10). Interestingly, the possible cellular mechanisms of the budding/trafficking of sEVs that could be used by HCV, HAV, HIV, Epstein-Barr virus (EBV) and Kaposi’s sarcoma-associated herpesvirus (KSHV) to spread from cell-to-cell viral elements or viral particles were recently revised [60]. Zhou et al. (2019), by using primary cultures of murine cortical neurons, showed that ZIKV uses sEVs as a mediator of viral transmission between neurons. Neuronal sEVs contained both ZIKV and viral RNA/protein(s) that were highly infectious to naïve cells. RNase A and neutralizing antibody assays suggested the presence of viral RNA/proteins inside EVs [65].

It is known that EVs may have different functions when interacting with other cells, modifying their naïve cellular behavior [66]. For monocytes (Figure 8), in the presence of different EVs samples, it was observed that naïve cells were differentiated and that they expressed a CD11b+ differentiation marker at high levels compared to the mock THP-1 cells and those stimulated with the mock C6/36 EVs. Likewise, the classical monocyte phenotype was changed to the CD14++ CD16+ intermediate phenotype with respect to the mock THP-1 cells and the mock C6/36 EVs stimulation assays (*p* < 0.0001 for CD14 and *p* < 0.01 for CD16). Intermediate and non-classical monocytes seem to be the main producers of pro-inflammatory mediators in response to viral infection [17,49]. The activation of naïve monocytes via the ZIKV C6/36 sEVs favors TNF-α mRNA expression, which suggests that ZIKV C6/36 EVs–human monocyte interplay plays a role in establishing a pro-inflammatory state. 

The possible participation of ZIKV C6/36 EVs in the infection and activation of vascular ECs was also evaluated. In cultures performed in HMEC-1 naïve cells in the presence of ZIKV (MOI 1), the mock C6/36 EVs, and the ZIKV small/large C6/36 EVs, the amount of ZIKV E protein was evaluated. Figure 10 shows the viral E protein presence at high levels in the ZIKV (MOI 1)-infected HMEC-1 cells as well as in the naïve ECs stimulated with ZIKV C6/36 EVs (RNase A + UV-treated and untreated). The E protein percentages were compared among values of all conditions against the mock HMEC-1 cells; these values were statistically significant (*p* < 0.0001) (Figure 10B,C). These data suggest that ZIKV-infected C6/36 EVs support the infection of mammalian host cells, including vascular ECs.

At the endothelial vascular cell level, our data indicate that ECs are susceptible to ZIKV infection and activation by ZIKV C6/36 EVs, and these EVs favor the induction of damage receptors, such as coagulation (TF) and inflammation (PAR-1) receptors, and adhesion molecule (ICAM-1) presence at the cell membrane’s surface level (Figure 11). Recent observations in human and animal models [51,52,67] suggest that, in severe Zika cases, different coagulation disorders occur. It has been shown that several viruses activate the coagulation system especially through TF receptor expression [53]. Dengue virus has also been shown to cause coagulation disorders in ECs [54].

Anfasa et al. (2019) provided in vitro evidence that ZIKV infection of human umbilical vein endothelial cells (HUVECs) induces apoptosis and increases TF production, which triggers the activation of secondary hemostasis [68]. Additionally, to evaluate the possible expression of a pro-inflammatory response by EVs released from ZIKV-infected C6/36 cells, we measured the TNF-α mRNA expression in naïve ECs infected by ZIKV (MOI 1) and those stimulated with the mock C6/36 EVs or the ZIKV C6/36 EVs (small/large isolates the same for RNase A + UV-treated). We found that, like ZIKV infection, the ZIKV C6/36 EVs were able to induce TNF-α mRNA expression in endothelial vascular cells (Figure 11D). The ZIKV C6/36 EVs also participated in vascular endothelial damage, with a weakening of the endothelial barrier integrity, and this was demonstrated using a Transwell assay (Figure 11E). We previously reported that DENV infection of ECs upregulates the TF coagulation receptor in endothelial vascular cells, which triggers the generation of hemostatic proteases (thrombin) favoring the activation of protease-activated receptors or PARs, which, in turn, induces signaling inflammatory pathways (via phosphorylation of MAPKs p38 and ERK1/2, by transcription of the NF-κB factor), thereby supporting the upregulation of adhesion VCAM-1 or pro-inflammatory molecules in ECs, being part of the pathogenic mechanisms for the vascular endothelial injury present in severe Dengue cases [54]. At present, it is under evaluation whether these signaling pathways can support the participation of ZIKV-infected vector cells in the activation and damage of vascular ECs that can contribute to the pathogenesis of severe ZIKV cases. The present data suggest that ZIKV C6/36 EVs allow for ZIKV elements (viral genome/protein) to modulate host response and enhance viral fitness abilities.

In summary, during ZIKV infection of C6/36 mosquito cells, small and large EVs were produced (Figure 12A). The mosquito EVs released from ZIKV-infected cells carried viral E protein and viral RNA and were able to infect naïve mosquito and mammalian cells. The ZIKV-infected mosquito EVs, then, modified the naïve cellular behavior, since they promote the infection, activation, and differentiation of human monocytes that favor a pro-inflammatory monocyte state (Figure 12B). At the endothelial vascular cell level, our data indicate that vascular ECs are susceptible to ZIKV activation and infection by C6/36 EVs, which favor the induction of tissue damage receptors, such as coagulation (TF) and inflammation (PAR-1) receptors, and adhesion molecule presence (ICAM-1) at the membrane surface level with an increase in cell permeability (Figure 12C). Knowledge of the targets’ cellular pathways that allow ZIKV to establish prolonged viral persistence could contribute to novel vaccines and therapies.

## Figures and Tables

**Figure 1 cells-09-00123-f001:**
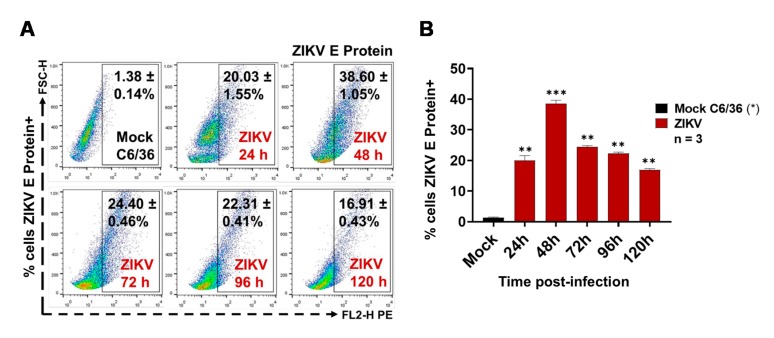
Zika virus (ZIKV) (multiplicity of infection (MOI) 1) infects C6/36 cells. (**A**) ZIKV envelope (E) protein detection at 24, 48, 72, 96, and 120 h post-infection (PI) by FACS assay. Dot plots are the representative mean ± standard deviation (SD) of the positive cells from three independent experiments. (**B**) ZIKV-infected cells percentages obtained by FACS. The ZIKV E protein levels were compared (by an unpaired Student’s t-test) with the mock C6/36 (*) value. Statistical significance was recognized as ** when *p* < 0.01, and *** when *p* < 0.0001.

**Figure 2 cells-09-00123-f002:**
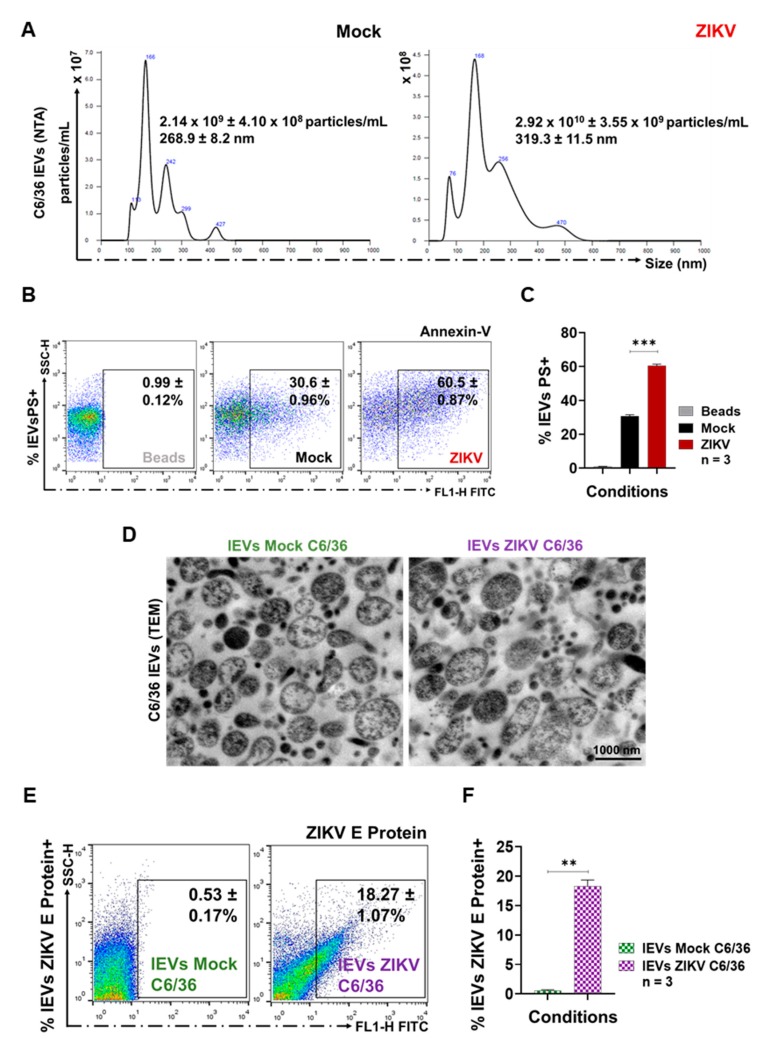
ZIKV-infected C6/36 cells issue the large extracellular vesicles (lEVs) phosphatidylserine (PS)+. (**A**) Nanoparticles tracking analysis (NTA) of the purified lEVs isolates from the mock and ZIKV-infected C6/36 cells. Histograms are the representative mean ± SD of the nanoparticle’s concentration (particles/mL) and the size (nm) from three independent experiments. (**B**) PS detection by the Annexin-V binding assay. The fluorescence emitted by Annexin-V binding is proportional to PS levels. FACS dot plots are the representative mean ± SD of the lEVs PS+ from three independent experiments. (**C**) lEVs PS+ percentages obtained by FACS. The PS levels were compared (by an unpaired Student’s t-test) with the mock lEVs (*) value. (**D**) Transmission electron microscopy (TEM) images from the mock C6/36 and the ZIKV-infected C6/36 lEVs (1000 nm scale). (**E**) ZIKV E protein detection on the lEVs surface by FACS assay. Dot plots are the representative mean ± SD of the lEVs ZIKV E protein+ from three independent experiments. (**F**) Percentages of the lEVs ZIKV E protein+ by FACS. The ZIKV E protein levels were compared (by an unpaired Student’s t-test) with the mock C6/36 (*) value. Statistical significance was recognized as ** when *p* < 0.01, and *** when *p* < 0.0001.

**Figure 3 cells-09-00123-f003:**
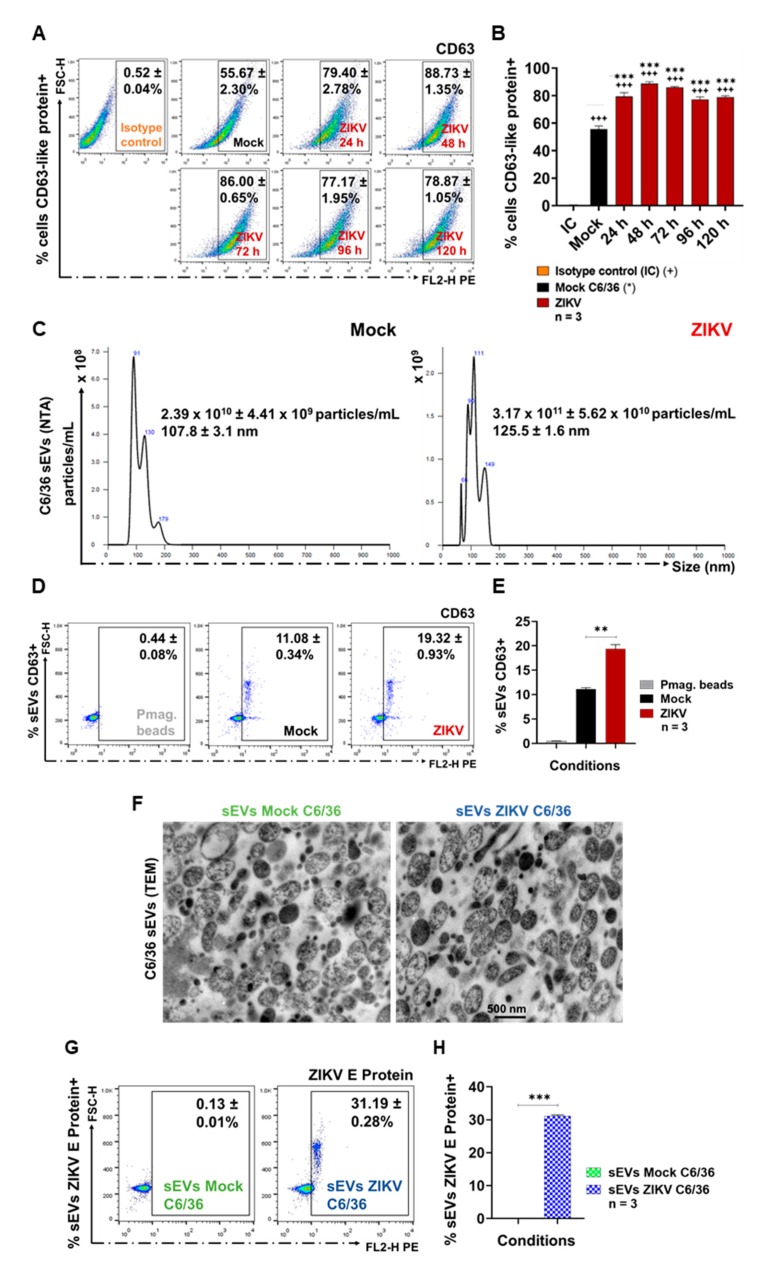
ZIKV-infected C6/36 cells issue small EVs (sEVs) CD63+. (**A**) CD63-like detection at 24, 48, 72, 96, and 120 h PI by FACS. Dot plots are the representative mean ± SD of positive cells from three independent experiments. Isotype IgG1 antibody was used as negative control. (**B**) Cells CD63-like+ percentages obtained by FACS. The levels of CD63-like protein were compared (by an unpaired Student’s t-test) with the isotype control (+) and the mock cells (*) values. (**C**) NTA of the purified sEVs isolates from the mock and the ZIKV-infected C6/36 cells. Histograms are the representative mean ± SD of the nanoparticle’s concentration (particles/mL) and the size (nm) from three independent experiments. (**D**) sEVs CD63+ coupled with paramagnetic bead detection by FACS. Dot plots are the representative mean ± SD of the sEVs CD63+ from three independent experiments. (**E**) sEVs CD63+ percentages obtained by FACS. The CD63 levels were compared (by an unpaired Student’s t-test) with the mock sEVs CD63+ (*) values. (**F**) Transmission electron microscopy (TEM) images from the mock and the ZIKV-infected C6/36 sEVs (500 nm scale). (**G**) ZIKV E protein detection on the sEVs coupled with paramagnetic beads by FACS assay. Dot plots are the representative mean ± SD of the positive sEVs ZIKV E protein+ from three independent experiments. (**H**) Percentages of the sEVs ZIKV E protein+ by FACS. The ZIKV E protein levels were compared (by an unpaired Student’s t-test) with the sEVs mock C6/36 (*) values. Statistical significance was recognized as ++ or ** when *p* < 0.01, and +++ or *** when *p* < 0.0001.

**Figure 4 cells-09-00123-f004:**
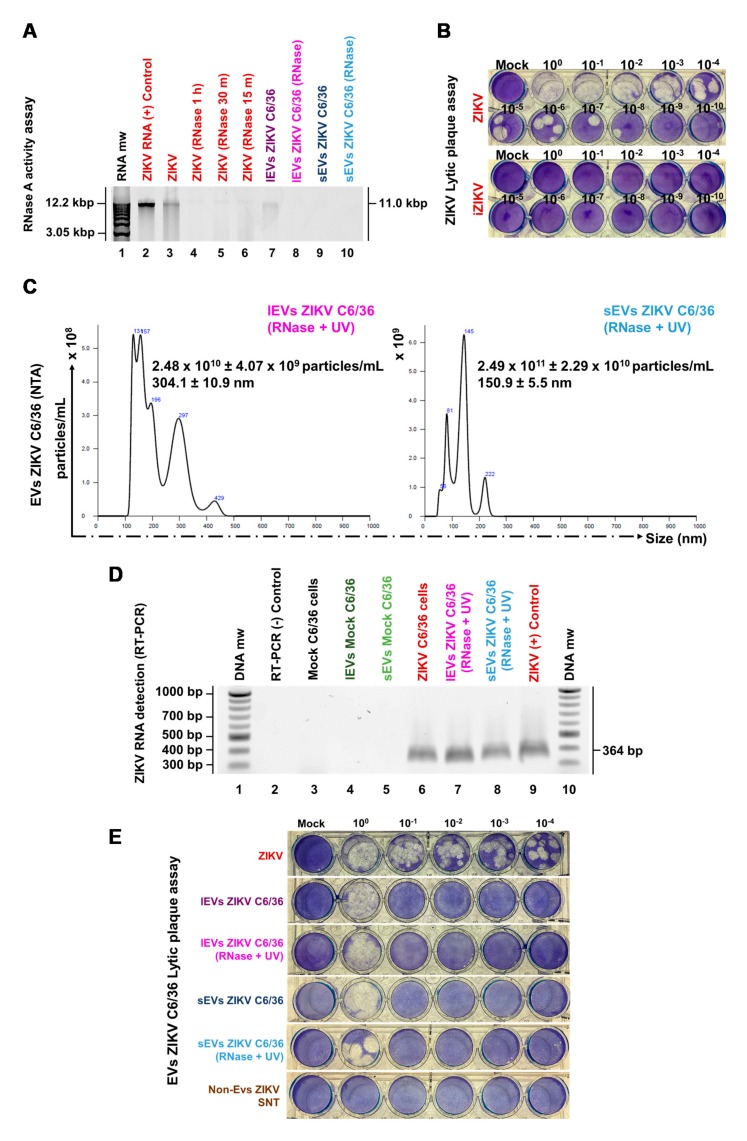
EVs from ZIKV-infected C6/36 cells, after ZIKV inactivation by RNase A activity assay and UV radiation, carry viral RNA and favor the mammalian cell infection. (**A**) RNase A activity assay. RNase A (10 µg/mL) was added to the purified ZIKV RNA and incubated for 1 h, 30 min, and 15 min at 37 °C with 5% CO_2_. Additionally, the RNase A activity (1 h incubation) was evaluated in C6/36 EVs isolates. The RNA degradation pattern was visualized on 2% ethidium bromide-stained 1.2% agarose gel. (**B**) Inactivated ZIKV (iZIKV) titration by a lytic plaque assay. (**C**) NTA of the EV isolates (from ZIKV-infected cells) treated with RNase A and UV. Histograms are the representative mean ± SD of the nanoparticle’s concentration (particles/mL) and the size (nm) from three independent experiments. (**D**) ZIKV RNA detection (RT-PCR) in ZIKV-infected C6/36 EVs (RNase A + UV-treated) samples. The ZIKV amplicon (364 bp from E genome conserved region) was visualized on 2% ethidium bromide-stained 1.2% agarose gel. (**E**) Evaluation of the ZIKV-infected C6/36 EVs (RNase A + UV-treated and untreated samples) in the viral transmission to naïve Vero cells by a lytic plaque assay.

**Figure 5 cells-09-00123-f005:**
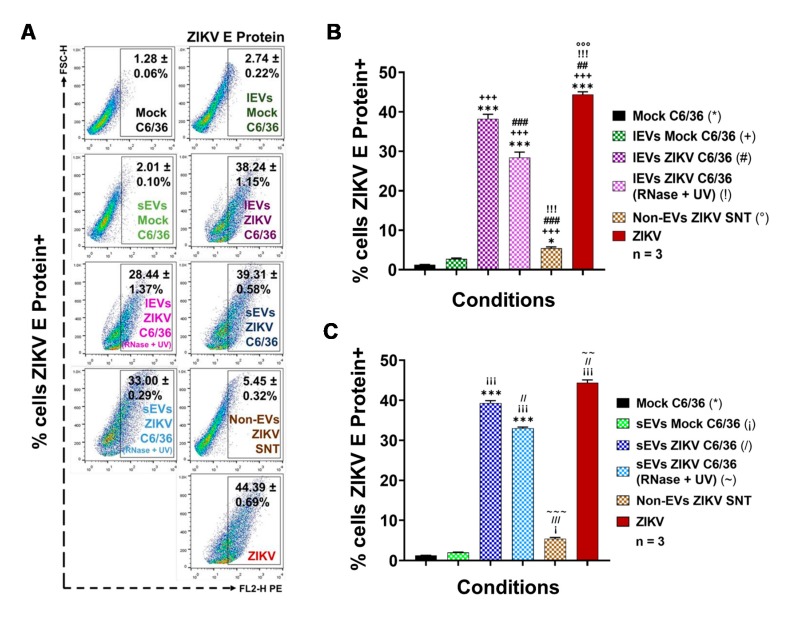
ZIKV E protein is present on the membrane’s surface of naïve C6/36 cells after the stimulus with ZIKV-infected C6/36 EVs. (**A**) ZIKV E protein detection at different EV stimuli conditions (FACS assay). Dot plots are the representative mean ± SD of the positive cells from three independent experiments. (**B**) Percentages of ZIKV E protein+ cells (FACS) after the lEVs stimuli. The levels of the ZIKV E protein were compared (by an unpaired Student’s t-test) between all conditions’ values. Statistical significance was recognized as *, +, #, !, or ° when *p* < 0.05, **, ++, ##, !!, or °° when *p* < 0.01, and ***, +++, ###, !!!, or °°° when *p* < 0.0001. (**C**) Percentages of ZIKV E protein+ cells (FACS) after the sEVs stimuli. The levels of the ZIKV E protein were compared (by an unpaired Student’s t-test) between all conditions’ values. Statistical significance was recognized as *, ¡,/, or ~ when *p* < 0.05, **, ¡¡,//, or ~~ when *p* < 0.01, and ***, ¡¡¡,///, or ~~~ when *p* < 0.0001.

**Figure 6 cells-09-00123-f006:**
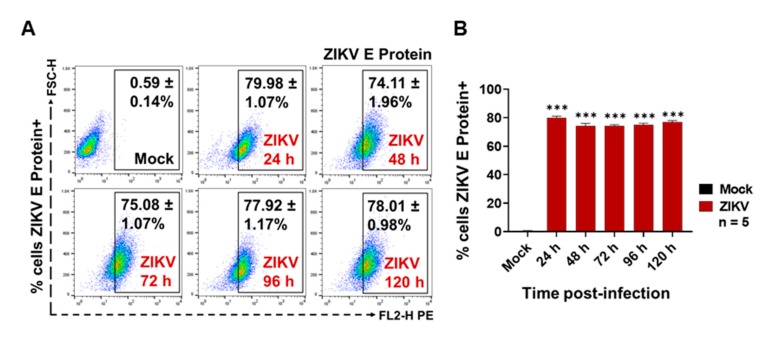
ZIKV (MOI 1) infects human monocytes. (**A**) ZIKV E protein detection at 24, 48, 72, 96, and 120 h PI by FACS assay. Dot plots are the representative mean ± SD of the positive cells from five independent experiments. (**B**) ZIKV-infected cells percentages obtained by FACS assay. The ZIKV E protein levels were compared (by an unpaired Student’s t-test) with mock C6/36 (*) values. Statistical significance was recognized as *** when *p* < 0.0001.

**Figure 7 cells-09-00123-f007:**
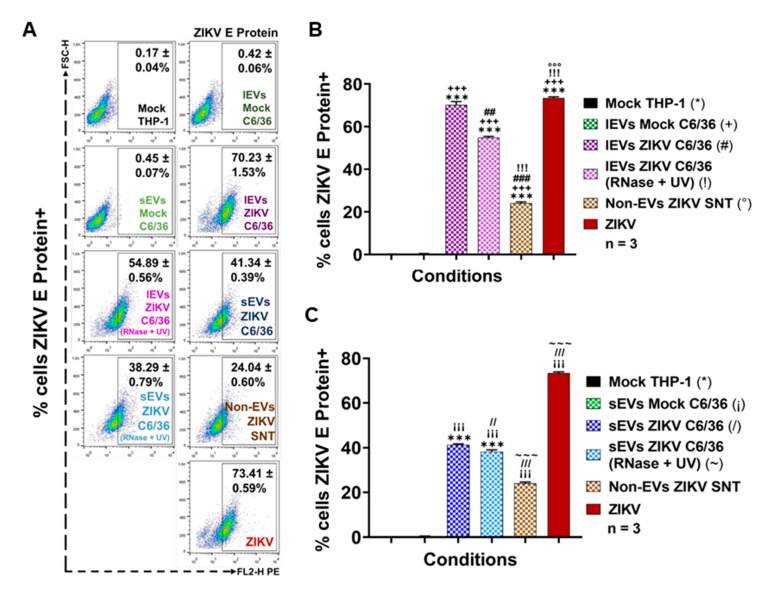
ZIKV E protein is present on the membrane’s surface of naïve monocytes (THP-1 cells) after the stimulus with ZIKV-infected C6/36 EVs. (**A**) ZIKV E protein detection at different EV stimuli conditions (FACS assay). Dot plots are the representative mean ± SD of the positive cells from three independent experiments. (**B**) Percentages of ZIKV E protein+ cells (FACS) after the lEVs stimuli. The levels of the ZIKV E protein were compared (by an unpaired Student’s t-test) between all conditions’ values. Statistical significance was recognized as *, +, #, !, or ° when *p* < 0.05, **, ++, ##, !!, or °° when *p* < 0.01, and ***, +++, ###, !!!, or °°° when *p* < 0.0001. (**C**) Percentages of ZIKV E protein+ cells (FACS) after the sEVs stimuli. The levels of the ZIKV E protein were compared (by an unpaired Student’s t-test) between all conditions’ values. Statistical significance was recognized as *, ¡,/, or ~ when *p* < 0.05, **, ¡¡,//, or ~~ when *p* < 0.01, and ***, ¡¡¡,///, or ~~~ when *p* < 0.0001.

**Figure 8 cells-09-00123-f008:**
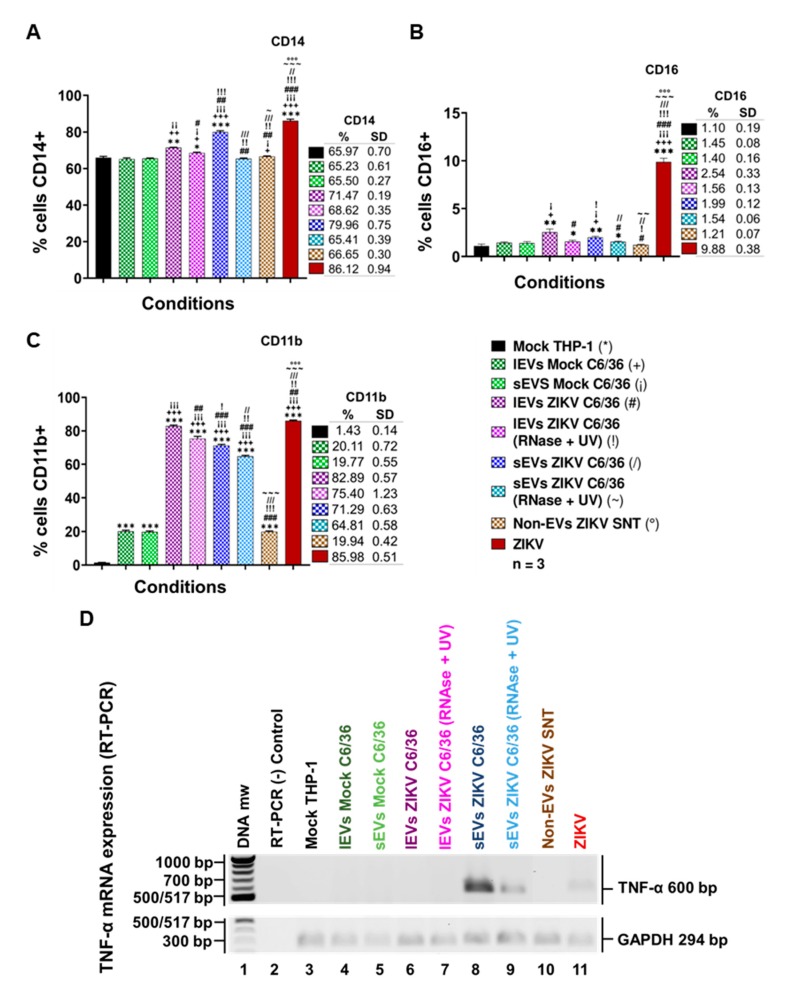
EVs from ZIKV-infected C6/36 favor the pro-inflammatory phenotype change in naïve human monocytes. (**A**) Monocytes CD14+ percentages (FACS) at different EV stimuli conditions from three independent experiments. (**B**) Monocytes CD16+ percentages (FACS) at different EVs stimuli conditions from three independent experiments. (**C**) Monocytes CD11b+ percentages (FACS) at different EVs stimuli conditions from three independent experiments. The CD14, CD16, or CD11b levels were compared (by an unpaired Student’s t-test) between all conditions’ values. Statistical significance was recognized as *, +, ¡, #, !,/, ~, or ° when *p* < 0.05, **, ++, ¡¡, ##, !!,//, ~~, or °° when *p* < 0.01, and ***, +++, ¡¡¡, ###, !!!,///, ~~~, or °°° when *p* < 0.0001. (**D**) Tumor necrosis factor-alpha (TNF-α) mRNA expression (RT-PCR) in naïve monocytes at different EVs stimuli conditions. The TNF-α genome conserved region (amplicon of 600 bp) was visualized on 2% ethidium bromide-stained 1.2% agarose gel.

**Figure 9 cells-09-00123-f009:**
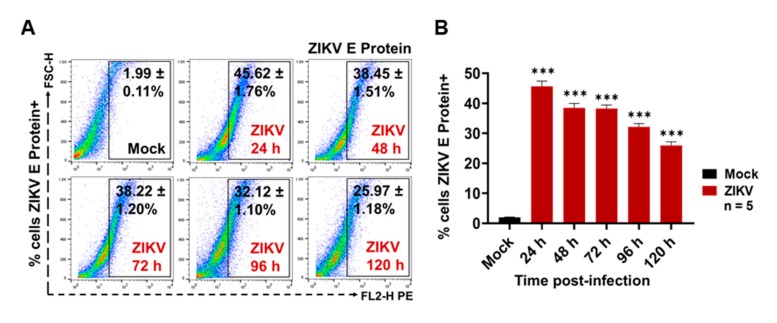
ZIKV (MOI 1) infects human endothelial (HMEC-1) cells. (**A**) ZIKV envelope (E) protein detection at 24, 48, 72, 96, and 120 h PI by the FACS assay. Dot plots are the representative mean ± SD of the positive cells from five independent experiments. (**B**) ZIKV-infected cells percentages obtained by FACS. The ZIKV E protein levels were compared (by an unpaired Student’s t-test) with the mock C6/36 (*) value. Statistical significance was recognized as * when *p* < 0.05, ** when *p* < 0.01, and *** when *p* < 0.0001.

**Figure 10 cells-09-00123-f010:**
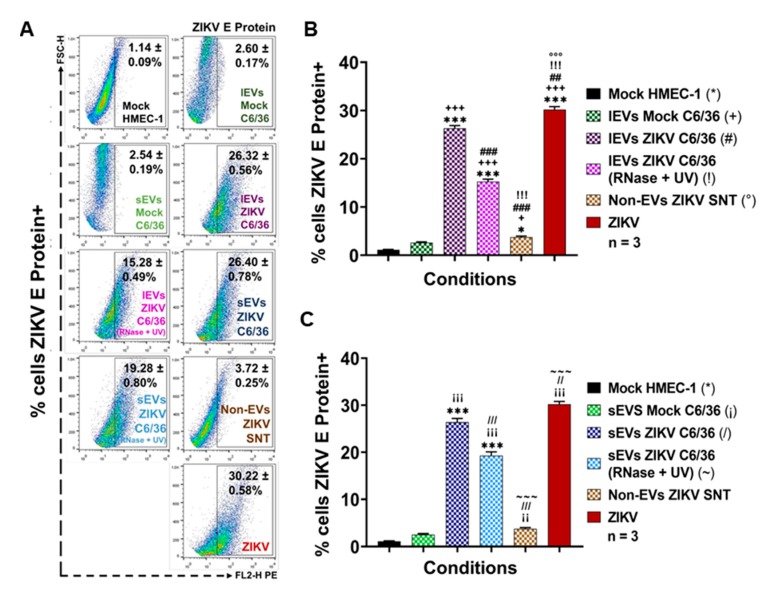
ZIKV E protein is present on the membrane’s surface of naïve endothelial cells (HMEC-1) after the stimulus with ZIKV-infected C6/36 EVs. (**A**) ZIKV E protein detection at different EVs stimuli conditions (FACS assay). Dot plots are the representative mean ± SD of the positive cells from three independent experiments. (**B**) Percentages of ZIKV E protein+ cells (FACS) after the lEVs stimuli. The levels of the ZIKV E protein were compared (by an unpaired Student’s t-test) between all conditions’ values. Statistical significance was recognized as *, +, #, !, or ° when *p* < 0.05, **, ++, ##, !!, or °° when *p* < 0.01, and ***, +++, ###, !!!, or °°° when *p* < 0.0001. (**C**) Percentages of ZIKV E protein+ cells (FACS) after the sEVs stimuli. The levels of the ZIKV E protein were compared (by an unpaired Student’s t-test) between all conditions’ values. Statistical significance was recognized as *, ¡,/, or ~ when *p* < 0.05, **, ¡¡,//, or ~~ when *p* < 0.01, and ***, ¡¡¡,///, or ~~~ when *p* < 0.0001.

**Figure 11 cells-09-00123-f011:**
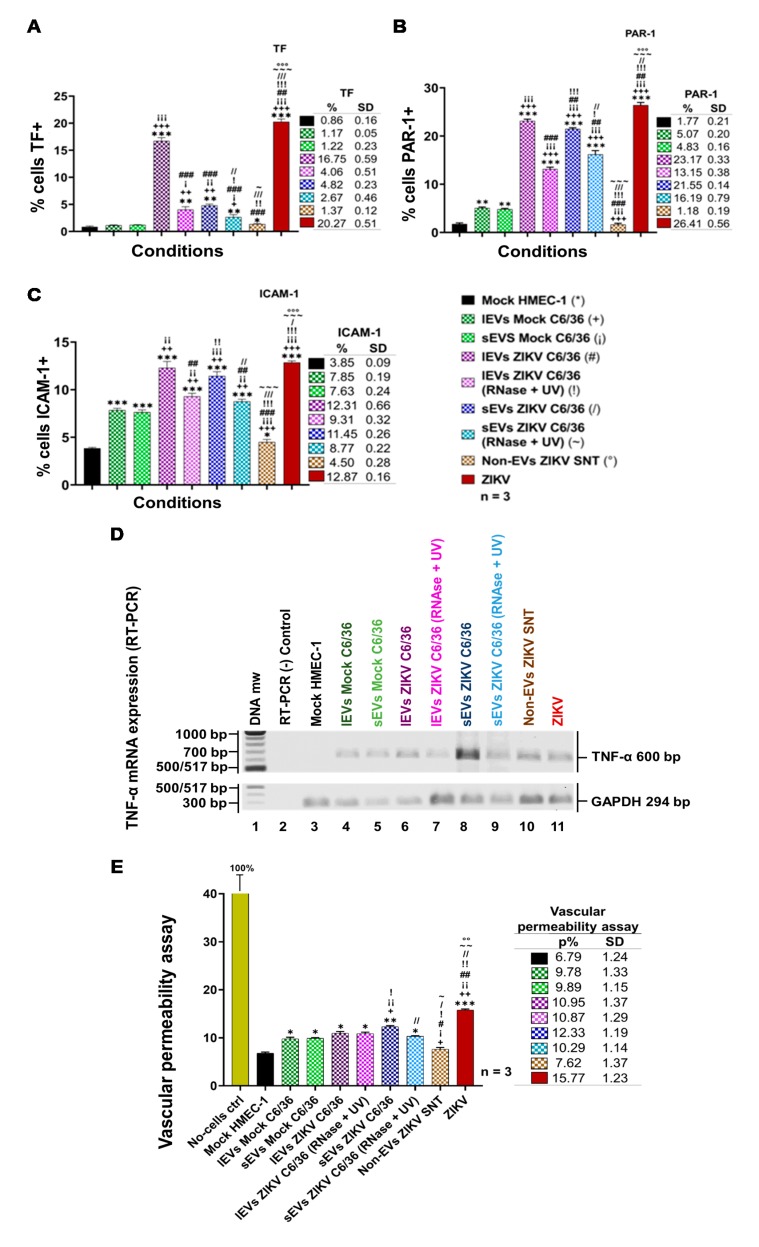
EVs from ZIKV-infected C6/36 modify towards a pro-coagulant, pro-inflammatory, and pro-adherent phenotype and favor permeability in naïve endothelial cells (ECs). (**A**) EC TF-1+ percentages (FACS) at different EVs stimuli conditions from three independent experiments. (**B**) EC Protease Activated Receptor+ (PAR-1) percentages at different EVs stimuli conditions from three independent experiments. (**C**) EC intercellular adhesion molecule-1+ (ICAM-1) percentages at different EVs stimuli conditions from three independent experiments. The TF, PAR-1, or ICAM-1 levels were compared (by an unpaired Student’s t-test) between all conditions’ values. (**D**) TNF-α mRNA expression (RT-PCR) in naïve ECs at different EVs stimuli conditions. The TNF-α genome conserved region (amplicon of 600 bp) was visualized on 2% ethidium bromide-stained 1.2% agarose gel. (**E**) Permeability percentages obtained by assessing the fluorescein isothiocyanate (FITC)-Dextran pass through the EC monolayers in the presence of different EV stimuli conditions. Three independent experiments were performed. For the 100% FITC-Dextran delivered control, a no-cell insert was used. The endothelial vascular permeability percentages were compared (by an unpaired Student’s t-test) between all conditions’ values. For all experiments, statistical significance was recognized as *, +, ¡, #, !,/, ~, or ° when *p* < 0.05, **, ++, ¡¡, ##, !!,//, ~~, or °° when *p* < 0.01, and ***, +++, ¡¡¡, ###, !!!,///, ~~~, or °°° when *p* < 0.0001.

**Figure 12 cells-09-00123-f012:**
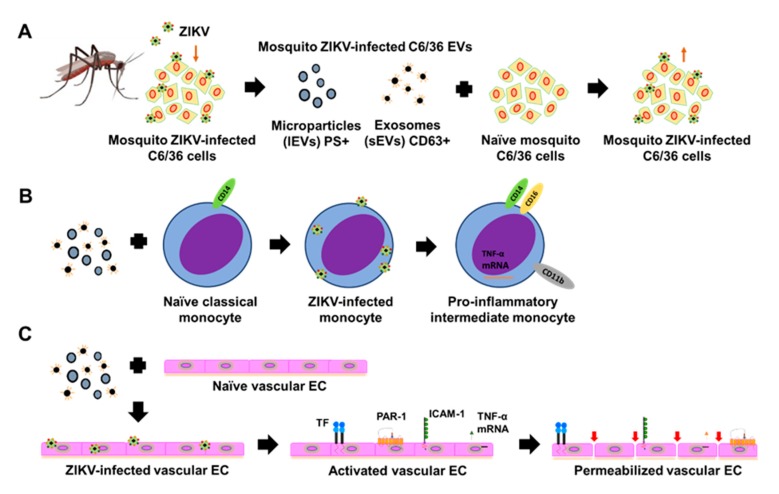
Extracellular vesicles (EVs) from ZIKV-infected mosquito (C6/36) cells participate in the modification of naïve cells’ behavior by mediating cell-to-cell transmission of viral elements (Graphic description). (**A**) ZIKV C6/36 EVs favor naïve mosquito cell infection. Mosquito image created with BioRender.com. (**B**) ZIKV C6/36 EVs promote infection and shift to a pro-inflammatory phenotype in monocytes. (**C**) ZIKV C6/36 EVs participate in the vascular EC infection and activation.

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
