# Peer review of "Participation of Extracellular Vesicles from Zika-Virus-Infected Mosquito Cells in the Modification of Naïve Cells’ Behavior by Mediating Cell-to-Cell Transmission of Viral Elements"

_cells, 2020, doi:10.3390/cells9010123_

Round 1

Reviewer 1 Report

The paper is an interesting work addressing the role of EVs in Zika disease and their capacity to participate to infectious process and pathology.  .

This paper demonstrated that EVS (small and large ones) issued from insect C6/36 cells that have been infected by ZIKV : 1) expressed the ZIKV protein E and carry some part of the RNA virus (only a segment of 364bp has been looked at); 2)Once this EVs in contact with naïve cells, part of the recipient cells will express the E protein 48h later; 3) sEVS and lEVs reproduce cell lysis in vitro; 4) EVs activated a proportion of monocytes and endothelial cells.

These different interesting results sign the fact that viral elements from the virus ZIKV are transmitted by EVs (small and large one), but do not support the fact that they transfer the virus as it is overstated in the summary, introduction, results and discussion (“EVs promote ZIKV transmission”, “viral spread”, “they transmit ZIKV to naïve cells”…). EVs issued from infected cells are reproducing part of the cytopathic effect induced by the mature infectious viral particles and part of the monocyte and endothelial cell activation.

The authors first demonstrated that the ZIKV infect insect cells (C6/36) as reported in fig 1. The infection is followed for 120h, with the formation of syncitia. The authors do not comment on the fact that there are less fluorescent cells after 48h of infection. Indeed, we could suspect that quantifying cells that have formed syncitia is not an easy task by FACS analysis ; moreover as the authors are talking of cytotoxicity, it would have be interesting to have followed the number of cells and putatively the apoptosis induced by the infection. Authors point that the protein E is expressed at the surface of the cells, however the quality of the immunofluorescent pictures (Fig1C) does not allow to conclude if there is also expression in the cytosol of the cells.

EVs produced by these infected cells are collected and directly enriched either via CD63 binding for small EVs and Annexin-V binding for large EVs. However, there is no characterization of the total small and large EVs as usually asked when working on EVs : expression of typical marker (TSG101, Alix, CD63,CD81) and lack of expression of marker that sign the presence of membrane debris (calnexin) by western-blot for example or at least the distribution of the size of the two different populations of EVs. Moreover, we would expect a demonstration that the CD63 bead selection of EVs is exempt of viral particles. Indeed, regarding the size of the Zika virus particles (50nm), we would expect to have them in the sEVS pellets.  A comparison of the sEVs pellet before and after the CD63 selection would be very informative in the efficacy of the CD63 beads to separate sEVS from viral particles. Figure 5D show that when the CD63 selected EVs are treated with RNAse and UV, there is less lysis in the well; could this be explain by the presence of a proportion of polluting viral particles? 

The only proof that we are dealing with EVs is the picture of EMT. It would be interesting to have a larger field of observation than the one presented, to confirm the heterogeneity of the different fractions of EVs. Did the authors observed viral particles into large EVs? 

Another very important point is that there is no quantification of the quantity of small and large EVs that are produced by the infected cells (quantity of protein of the pellets? DLS or nanosite quantification ? etc..) and that are used for functional test. We only know that when doing the functional test, the authors are using 250ul of the suspension of the different EVS. We don’t know which quantity of supernatant is processed, and in which quantity of solution the different pellets of EVS are resuspended. It is necessary to make sure that the same quantity of vesicles (small and large) were put in contact with naïve cells to be able to compare they effects. Moreover authors have chosen to collect small or large Evs at different time post-infection which is concerning when you want to compare they effects.

Line 104-105 : overstatement : the data presented in this paper do not support that “EVs allow ZIKV to infect mammalian cells via receptor-independent mechanisms”. EVs can enter cells through interaction ligand-receptor.

Line 131  : “ZIKV from Vero cells was used in all the experiments”. Authors could clarify why they used virus issued from mammalian cells, while this virus has been previously amplified in C6/36 cells ? indeed, it is surprising to compare the effect of a virus “made” by a mammalian cells to EVs “made” by an arthropod cell. The ideal would have been to compare the virus and the EVs issued from a same species.

An important control is missing in the different functional experiment : the virus ZIKV treated with RNase and UV. Indeed, once the virus production treated, do we still observe cell lysis ? activation of monocytes ? Activation of endothelial cells? This control would help us to better decipher the different capacity of an active viral particle and other type of vesicles carrying/expressing viral elements.

This paper is a very long manuscript that would require to be written in a more synthetic way. There is lot of redundancy in the writing between the introduction, methods, results and conclusion parts. This paper proposed 15 figures, the last one being a schematic of EVS impact on the monocytes and the endothelial cells. Some of the figures should be combined and others added as supplementary data. The Y legend of the graph should be simplified as the percentage of positive cells (and not the quantity of E protein as overstated in the paper). The gels are very difficult to read. A black and white version, with a better contrast would help a lot.

Once the authors have explained that small EVs are called sEVs or exosomes, and that large Evs are called lEVs or microparticles or microvesicles, the authors should choose one name ad only use it till the end of the manuscript.

The term “co-culture” is not adequately used all along the paper. Co-culture would be adequate for the culture of 2 different type of cells, which is not the case here. The EVs are not co-cultivated with the recipient cells (EVs do no replicate as cells). Instead the cells are “treated with the EVs” or “exposed to EVs”.

Author Response

      "PLease see the attachment"

Reviewer 2 Report

Martinez-Rojas, P.P. et. al. systematically studied the extracellular vesicles from ZIKV infected mosquito cells carries the essential pieces of viral RNAs, retain the capacity of cell-to-cell transformation of naïve cells, cultured human monocytes, naive endothelial cells, change the cell behavior of the infected cells and promote the proinflammatory and pro-coagulant states. The data were well documented and the experimental logic was solid. The conclusion appears well supported by the data. There are some concerns that need addressing before final publication. I am listing them in the following. 1) For tittering of EVs, can droplet digital PCR be used for measuring the copy-number of ZIKV E genome RNA? 2) IN Fig. 1, can a control inactivated virus (or viroid) be used as negative control for ZIKV? Can a control antibody against cell surface protein be used as control for specificity? Can the morphological changes of cells be quantitatively compared among cell population in each condition? 3) Fig. 2B, why lEVs from Mock cells show significant PS+ in the outer leaflet? Isn’t the PS asymmetry preserved in the vesicles? The gate for Fig. 2C should be extended leftward a bit to cut at the 3% tail of the BEADs peak. The gate position might change the statistical difference between the mock and ZIKV infected cells. Fig. 2D, the MPs in both conditions seem to be as large as 1200 nm in diameter. It is better to show a histogram of MP diameter in both conditions from the EM images. 4) Fig. 3: an inactivated virus or viroid as negative control would be better. 5) Fig. 3C, why the mock treated cells have high expression of CD63-like proteins than the isotype control? 6) Line 431 (pg 11), why only internalization is considered? How about budding and formation of EVs? 7) Line 440 (pg 11), whati s “previous stain”? 8) Fig. 4E, a histogram of sEV diameter is needed to show the size distribution. The data show that the sEVs are mostly larger than 200 nm in diameter with the largest ones approaching 560 nm. These are different from the classification stated in the text: MP >200 nm and sEV < 200 nm. The data are not in accord with the statement in line 446-447 about sEV dimensions. 9) Data in Fig. 5B do not support the conclusion on the protection of RNAs from RNAse. 10) Fig. 5B/5C, need controls of EVs (detergents, UV, RNAse) to test whether permeabilization of EVs can allow full digestion of RNAs by RNAse A. 11) Fig. 8B and Fig. 11B show a valley at 48 hr PI, different from Fig.3B. Why? 12) Fig. 8C, the fractions of morphologically changed cells seem to be smaller than the data in Fig. 8A/8B. Can these cells be quantified? 13) Fig. 10E, why TNF-alpha transcripts are much higher for the sEV treated cells? Are the lEVs and sEVs having the same likelihood of getting into the cells? 14) Fig. 11C, the cytopathic effects need quantification from many cells, instead of showing typical images. 15) Fig. 13E, the same question as #13. 16) Fig. 13A vs 13E, why the TF signal of LEVs are much stronger than other vesicle samples, while the TNFalpha is much higher in sEV treated cells? The two are not coupled?

Reviewer 3 Report

This study examines the ability of extracellular vesicles (EVs) from Zika virus-infected mosquito cells to transmit infection to other cells. The authors parallel this study with the recently reported work with dengue virus showing transmission through EVs. 

With 14 data figures, the authors conduct an extensive study using several cell types.  For most experiments, flow cytometry and microscopy are used to show results from two methods.  Ultimately, the conclusion is that EVs participate in Zika virus transmission, which in turn, affects cells (pro-inflammatory state, permeabilization, etc). 

Overall, the manuscript needs to be edited for style.  If the manuscript is re-written and the below comments are addressed, then it would improve the quality of the article. 

Introduction: the content seems appropriate, but the style needs work.  For example, Zika was not recognized in 2016.  The virus was discovered decades ago.  Since 2007 there has been a shift in the epidemiology of Zika with human cases increasing.  A ZIKV epidemic in Brazil began in 2015 and spread rapidly throughout South and Central America in 2016.   The authors know this, but the appropriate message doesn't come through in the writing. 

Methods: 

1.  "Vero" is the commonly used cell designation in the literature and ATCC (not "VERO").

2. Using "lEV" to designate large EVs is too easy to confuse with "iEV" because the lowercase l (L) and uppercase I (i) look the same.  Suggest to change it to "LEV" using all capital letters (and then to use "SEV" for small EVs) or perhaps to "lgEV" and "smEV. " 

3. The true passage history of MR766 is not reflected in this section.  MR766 was passaged over 100 times since its isolation in 1947.  Perhaps the authors mean that it was passaged twice in C6/36 in their lab.  This section needs to be cleared up.  Also, what was used for the experiments?  The section states Vero P3, but that cannot be when the previous statement explains the strain was passaged twice in C6/36 cells.

4. When were plaque assays fixed?  The manuscript states when CPE were observed, but how long after incubation is this?  Typically methods state the number of days post-infection.

5. The section on flow cytometry and IF staining should be separated into two sections.  How were the various cells harvested for flow cytometry?  Were they trypsinized, scraped, harvested through vigorous pipetting?  Instead of using IF to detect internal E-protein, why wasn't the flow cytometry procedure for staining conducted after permeabilization of cells to compare the expression?  

6.  The section on isolation of EVs needs its own flowchart or schematic.  At each step the authors should indicate what each fraction (pellet or supernatant) contains and how it is checked or monitored.  This information is not clear and is the most important aspect of the research described.  Without a clear explanation of these steps, the reader cannot know or follow the work. It is difficult to follow what the 'control' fraction is.  How much culture is used at the beginning prior to harvest and isolation?

7.  Line 274: how many cells were seeded per well?  1x10^6 or 2.5x10^5?  Perhaps section 2.14 could use a schematic to make the text less confusing.

Results: Without more knowledge regarding the LEV and SEV fractions, it is difficult to evaluate the results.

1 What are the replicates in these experiments?  Were LEV and SEV fractions isolated three times from three different infections?  Or was there only one preparation of EVs used in three different transmission experiments for C6/36, HMEC, etc?

2.  Although the authors show that the data are statistically significant throughout the manuscript, it is not clear how the infection % difference between 28% and 38% (for example) is biologically relevant or significant.  This is the case throughout the manuscript. 

3. Are the authors implying that ZIKV infection between C6/36 cells occurs mainly (or really only) through EVs?  This would indicate that infective "free" virus is not released from C6/36 cells and that almost all infection occurring in culture is through vesicles.

4.  Why does infection level (E protein expression) of THP-1 and HMEC decrease from 24 to 48 hr then increase again at 72 hr?

Discussion:

The manuscript contains a lot of data and information.  However, there is little interpretation of the different percentages of E protein in LEVs and SEVs and in the different cell types.  

Round 2

Reviewer 1 Report

The format of this proposition is still closer to a thesis manuscript than a scientific article : 12 figures and 14 supplementary data. The authors should better synthetize the resulst and the writting to fit to a scientific article format.

Author Response

We would like to thank you for your helpful observations, which have substantially

contributed to improve the quality of the present manuscript.

The extensive editing of English language was performed.

Reviewer 2 Report

My concerns were addressed. But the English expression needs improving. The response letter was difficult to understand in a few places. The same is for the manuscript. It needs extensive revision. I am giving a few examples for page 1.

The title reads awkward. It might be better to change into one that is easy to understand. For example: “extracellular .. in the modification of naïve cells’ behavior by mediating cell-to-cell transmission of viral elements”

Line 19, “For the control of Zika” might be better.

Ln 21: the “targeted …” is better. Not “the targets ..”

Ln 22: “contributes” not “contribute”

Ln 25/26, “EVs …carry .. and are able to …”  is more accurate.

 Ln26, change “favor” to “promote”

Ln27: “and induce” may be better than “inducing”.

            It is not clear if the pro-inflammatroy state promotes differentiation.

Ln 28: “participate …. Damage of endothelial vascular cells by inducing … at the endothelial surface of the cell membranes, and promote a …”

Ln 38. “ZIKV was first isolated in 1947 from …” is better.

Ln 39 , “in Uganda. Then again in 1949, ZIKV was

Author Response

We would like to thank you for your helpful observations,which have subtantially

contributed to improve the present manuscript. 

Extensive editing of English language was performed.

Reviewer 3 Report

The authors thoroughly addressed all of the comments and concerns raised during the initial review. Additional edits, figures, and schematics have strengthened the manuscript.

Author Response

We would like to thank you for your helpful observations, which have substantially

contributed to improve the quality of the present manuscript. An extensive editing

of English language was performed.